# Unsupervised learning of perceptual feature combinations

**Minija Tamosiunaite**[1,2]*, **Christian Tetzlaff**[3,4], **Florentin Wörgötter**[1]

**1** Department for Computational Neuroscience, Third Physics Institute, University of Göttingen, Göttingen, Germany, **2** Vytautas Magnus University, Faculty of Informatics, Kaunas, Lithuania, **3** Computational Synaptic Physiology, Department for Neuro- and Sensory Physiology, University Medical Center Göttingen, Göttingen, Germany, **4** Campus Institute Data Science, Göttingen, Germany

\* minija.tamosiunaite@vdu.lt

**Data Availability Statement:** All relevant data are within the manuscript and its Supporting information files.

**Funding:** Supported by the German Science Foundation (DFG), grant WO 388/17-1 (F.W.) and grant TE 1172/7-1 (C.T.), as well as by the

## Abstract

In many situations it is behaviorally relevant for an animal to respond to co-occurrences of perceptual, possibly polymodal features, while these features alone may have no importance. Thus, it is crucial for animals to learn such feature combinations in spite of the fact that they may occur with variable intensity and occurrence frequency. Here, we present a novel unsupervised learning mechanism that is largely independent of these contingencies and allows neurons in a network to achieve specificity for different feature combinations. This is achieved by a novel correlation-based (Hebbian) learning rule, which allows for linear weight growth and which is combined with a mechanism for gradually reducing the learning rate as soon as the neuron's response becomes feature combination specific. In a set of control experiments, we show that other existing advanced learning rules cannot satisfactorily form ordered multi-feature representations. In addition, we show that networks, which use this type of learning always stabilize and converge to subsets of neurons with different feature-combination specificity. Neurons with this property may, thus, serve as an initial stage for the processing of ecologically relevant real world situations for an animal.

## Author summary

During foraging and exploration, the neural system of animals is flooded with numerous sensory features. From this confusing signal repertoire, it needs to learn extracting relevant events often encoded by specific perceptual feature combinations. For example, a specific smell and some distinct visual attribute may be meaningful when occurring together, while by themselves these features are irrelevant. Learning this is complicated by the fact sensory signals occur with different intensity and occurrence frequency beyond the control by the animal. Here we show that it is possible to train neurons with external signals in an unsupervised way to learn responding specifically to different feature combinations largely unaffected by such presentation contingencies. This is achieved by a novel learning rule which achieves stable neuronal responses in a simple way by gradually reducing the learning rate at its synapses as soon as the neuron's response to the feature combination exceeds a certain level. This allows neurons in a network to code for different

European Commission H2020, grant no.: 899265 "ADOPD" (F.W.) and the German Federal Ministry of Education and Research (BMBF) grant no. 01IS22093A "KISSKI" (C.T.). The funders had no role in study design, data collection and analysis, decision to publish, or preparation of the manuscript.

**Competing interests:** The authors have declared that no competing interests exist.

feature combinations and may facilitate ecologically meaningful evaluation of perceived situations by the animal.

## Introduction

Coincident events or features can be highly relevant for animals and humans, and recognizing feature combinations may make all the difference between danger and safety. The red color of a mushroom paired with white surface dots as compared to a red one with a plain surface makes the difference between the poisonous Amanita muscaria (toadstool) and the eatable Amanita caesarea. While humans learn such feature combinations usually by supervision, animals often do so via trial and error. For example, rats and other animals perform scouting and probing of novel food sources until found to be safe. Repeated exposure to sensor-perceivable feature combinations in conjunction with no negative effects will then lead to the conclusion that it should be safe to eat this.

A central problem that arises here is that features will not only occur in combination but also on their own. This will happen with different individual- as well as coincidence-occurrence frequencies and, in addition, usually also with variable intensity. Thus, in order to learn the meaning of combined features, the nervous system must learn this without being distracted by this variability. While supervised learning methods, like LMS algorithm [1] could address this problem in efficient way, here we investigate unsupervised learning. The latter is much simpler from the point of view of biological implementation, as it does not require additional evaluative sub-systems and mechanisms. As shown below, with unsupervised learning one can detect feature combinations already at the level of a single neuron. To achieve this, neuronal plasticity must come to a halt as soon as a combination has been recognized, otherwise ongoing plasticity would lead to undesired responses to individual features.

Such an ecologically driven stopping of learning is a non-trivial problem for unsupervised learning, though. For example, Hebbian learning leads to unbounded (divergent) weight growth. Many stabilization methods and/or augmentations of the original Hebbian learning rule have been suggested to prevent this, for example Oja's rule [2], the Bienenstock, Cooper, Munro rule (BCM, [3]), subtractive normalization methods [4] and several more. More recently, learning rules had been introduced which combine Hebbian weight growth with a homeostatic, balancing term, called synaptic scaling [5–7], for achieving convergence to a target activity [8]. However, below we will show that these methods cannot reliably address the problem of differentiating cases with coincidences of two or more features.

As a consequence the issue of how to control weight development in an unsupervised way such that a neuron will reliably code for multiple-feature combinations remains unresolved. Here we suggest a rather simple solution to this. When growing weights in a network, combination-selective responses can be achieved by gradually dropping the learning rate to zero (simulated annealing) as soon as the neuron's activity is getting "large enough", which happens earlier for combined than for individual stimuli. Before describing details, this mechanisms can best be understood by an example (Fig 1). Here two similar inputs (A,B) had been presented randomly with some coincidence between them (vertical dashed lines). Due to learning, the neuron's output (C) gradually gets bigger, where the response amplitudes that occur for coincident inputs will always exceed individual ones. When a response passes the annealing threshold the learning rate is reduced (red curve in D) and after some time, when the rate has dropped to zero, weight growth will come to a standstill. This mechanism keeps the individual

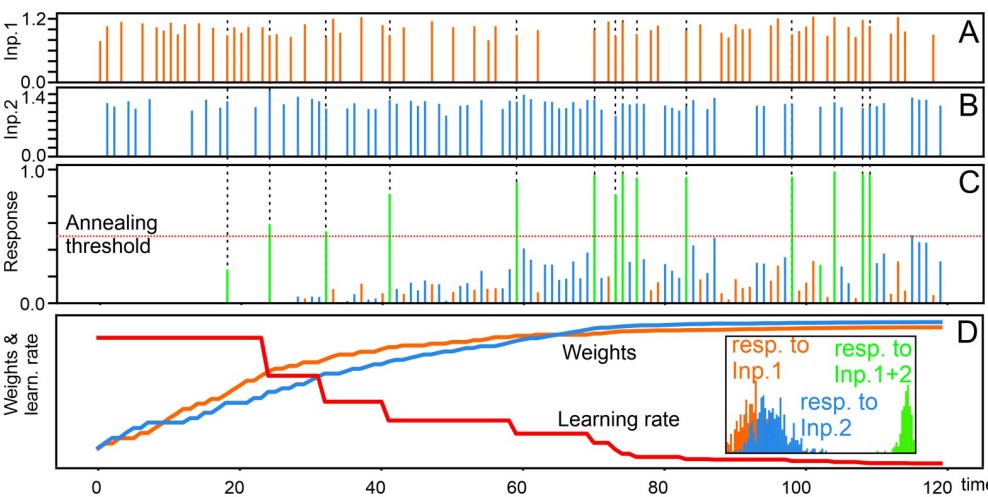

**Fig 1.** Exemplary development of the response (C), synaptic weights (D), and annealing characteristic (D, red) for a neuron with two inputs (A,B) with mean amplitudes of 1 and 1.2, respectively, and same average occurrence frequency; dashed lines show input coincidences. Inset shows the finally resulting distribution of neural responses between an activation of zero (left) and one (right).

responses small as compared to the combined ones and the resulting output distributions (inset) remain separate with a large gap between single and coincident responses.

Simulated annealing has become a textbook method in reinforcement learning (RL), for example for step-size reduction [9] or for reducing exploration rates [10] as well as in deep-RL [11]. Annealing is also widely used in supervised learning [12–14] as well as in different variants of Hebbian learning [15–18], the latter being most closely related to the investigations in this study. However, annealing in those studies is applied as an additional mechanism to ensure an efficient convergence of weights, while we are not aware of studies which would analyze annealing as the main factor for activity stabilization on its own.

Central to our approach is that the principle of using the neuron's output as the determining variable for the annealing leads to the advantageous property that neurons in a network will indeed develop specificity for different input (or feature) combinations. This differs from mere spike-coincidence detection because—as discussed above—the learning of input combination specificity needs to be independent (within reason) of the intensity of the input, represented by its occurrence frequency and its amplitude (or input firing rate). Amplitude invariance can to some degree be achieved using network-intrinsic normalization (e.g. [19]) by which differently strong activity, e.g. from external sensory features that converges onto a cell, will still lead to similar, albeit not identical, responses. These could then serve as the normalized inputs to the learning neuron. Another aspect that leads to problems is that learning needs repetitions. However, the brain has little or no influence on the occurrence frequency of any external stimulus or stimulus combination. Hence, to reliably learn input combination specificity the system must tolerate quite some variability in the occurrence frequencies of the different inputs as well as concerning their coincidences.

The central contribution of this study is showing that the annealing mechanism allows reliably encoding coincident feature combinations of two or more features in spite of amplitude and frequency variations of the input signals. This is achieved without having to adjust the neuron's parameters for different stimulus situations. In addition, we show that—for multiple inputs—the neurons' output distributions are ordered by the total (average) input intensities.

This is another factor, which could be ecologically relevant as those neurons this way represent quite faithfully "what comes in from the environment".

We start this investigation in the first part of the paper by analysing a simple case of a neuron with only two inputs and compare our rule to other, conventional learning mechanisms. Then we address the aspect of multi-input ordering. We show that other rules fail to achieve these properties and provide also a detailed analysis of the BCM rule, which could be seen as a contender to our approach. This is then extended to a recurrent network to address the issue of multiple coincidences. We finally discuss possible biological mechanisms that might support this function and also other issues concerning the learning of input coincidences.

## Materials and methods

In this section we will first describe our neuron model, then the learning rule that we are proposing. Afterwards we briefly specify the traditional learning rules to which we are comparing the newly proposed method. Finally, we describe a setup, where we have embedded this rule in a recurrent neural network.

### Neuron model

To obtain the neuronal response, first we calculate the weighted sum of the inputs:

$$y = \boldsymbol{\omega}^T \boldsymbol{u}, \tag{1}$$

where $\mathbf{u} = (u_1, \ldots, u_n)^T$ are inputs, $\boldsymbol{\omega} = (\omega_1, \ldots, \omega_n)^T$ are weights, and $n$ is the number of inputs. In analogy to real neurons, we will call $y$ the *membrane potential*. We will first analyze the simplest neuron that can detect co-incidences with $n = 2$, but will increase the number of inputs in the later-shown recurrent network example. To calculate the actual neuronal response $v$, called *spike rate or rate*, we apply a nonlinear function:

$$v = f_s(y) = \begin{cases} \frac{1}{0.9}\left( \frac{1}{1+e^{-b(y-0.5)}} - 0.1 \right) & , \text{if} \geq 0, \\ 0 & , \text{otherwise.} \end{cases} \tag{2}$$

where $b = 10$ if not indicated differently. Coefficients were set to obtain the response characteristic shown in Fig 2A. This represents a sigmoidal function with a threshold $y_T \approx 0.281$, beneath which the firing rate will be zero:

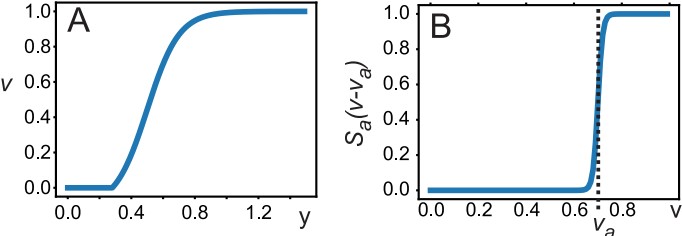

**Fig 2. Functions used in model equations.** A: Neural activation, see Eq (2); we use a saturating function, where in case of small membrane potential ($y$) the activation $v$ is zero, which is based on empirical observations, e.g., see. [20, 21]; B: Annealing function, which renders a close to zero annealing rate until threshold value $v_a$ is reached, and afterwards increases abruptly, see Eqs (4) and (5); note that this function is additionally scaled by the annealing rate $\rho$ in Eq (4).

### The Hebb rule

We investigate the effect of learning rate annealing on the Hebb rule given by:

$$\frac{d\boldsymbol{\omega}}{dt} = \mu(t)\boldsymbol{u}g(y), \tag{3}$$

with **u** the input vector of the neuron, $g(y)$ the influence of the neuron's output on the learning and $\mu(t)$ the learning rate, which will change over time due to annealing.

### Learning rate annealing

Central to our method, however, is that the spike rate $v$ guides the annealing of the learning rate $\mu(t)$, where we start annealing, as soon as the neuron has reached "high enough" outputs $v$. The annealing equation is as follows:

$$\frac{d\mu}{dt} = -\rho S_a(v - v_a)\mu, \tag{4}$$

where $\rho$ is a rate factor, $v_a$ the annealing threshold, and $S_a(x)$ is another sigmoidal function:

$$S_a(x) = \frac{1}{1 + e^{-\beta x}}, \tag{5}$$

where we used $\beta = 100$ to obtain a steep step-like transition (see Fig 2B). However, the method will work in a similar way with several times bigger or smaller $\beta$. Learning starts at $t = 0$ with $\mu(0) = \mu_0$. The sigmoidal function leads to the following effect: at the time when $v$ exceeds $v_a$, the annealing rate abruptly increases. The value for $v_a$ is expected to be around or higher than the inflection point of $v$. We have investigated $v_a \geq 0.45$. Note that, if annealing happens too early, the neuron's differentiation capability remains low, as its activation function non-linearity will not play any role.

### Hebbian learning rules with annealing

We define for Eq 3 different characteristics for $g$. First, there is a rule, which we call *annealed membrane Hebb (AMH)* rule, defining $g(y) = y$, hence:

$$\frac{d\boldsymbol{\omega}}{dt} = \mu(t)\boldsymbol{u}y \tag{6}$$

This rule leads to exponential weight growth (see Eq 15) due to the fact that the neuronal output coupled with the learning-equation creates a positive feedback loop.

To avoid this problem, we have replaced this rule with one that is largely output independent and leads to linear weight growth (see Eq 13). This so-called *annealed Linear Learning (ALL)* rule uses $g(y) = H(y - \eta)$ with $H$ being the Heaviside function:

$$H(y - \eta) = \begin{cases} 1 & \text{, if } y - \eta > 0, \\ 0 & \text{, otherwise.} \end{cases} \tag{7}$$

Hence, the ALL-rule is given by:

$$\frac{d\boldsymbol{\omega}}{dt} = \mu(t)\boldsymbol{u}H(y - \eta) \tag{8}$$

This learning rule augments traditional Hebbian learning by the assumption that weight change will *not* depend on the actual activation of the neuron. Instead learning will start as

soon as the membrane potential $y$ exceeds a threshold $\eta$ and then only depends on the incoming input(s). Analysis of the experimental literature shows (see Discussion) that—especially at dendritic spines—this type of learning may be biophysically more realistic than other variants of the Hebb rule.

Below, we will also show that the ALL rule works best for the here-investigated task of coincidence detection. For simplicity, we used $\eta = 0$ but results will not change much as long as one uses reasonably small values for $\eta$. Note, that the weight update routine described above in case of $\eta = 0$ holds some similarity to Rosenblatt's perceptron learning rule [22]. However, different from Rosenblatt's perceptron, where knowledge on the desired outputs is assumed and error terms are used, we analyze unsupervised learning, where self-organization happens without supplying knowledge about the desired output of the neuron. For more considerations on supervised vs. unsupervised learning see Discussion section, subsection "Comparing to other learning principles".

Note that, in principle, one can also define a Hebb rule that relies on the actual rate $v$ and, hence, considers the output transform (Eq 2) by setting $g(y) = v$. However, this case, which is governed by the sigmoid output function of the neuron, can be tuned to either approximate the annealed membrane Hebb (AMH) or the annealed linear learning (ALL) rule. Hence, we will not consider it any further.

## Reference models

We compared our method to the BCM-rule [3] and the Oja-rule [2] as well as to a newer approach called synaptic scaling [8].

For BCM, there exist several linear as well as non-linear versions in the literature (e.g. [3, 23, 24]). We had analyzed these rules, but here we show results only for the (non-linear) formulation introduced by Intrator and Cooper [23], which superseded the others for the here-investigated tasks. However, in the Results section we will also briefly discuss results from the other BCM rules.

The Intrator-Cooper BCM rule is given by (see [25]):

$$\frac{d\boldsymbol{\omega}}{dt} = \mu v(v - \Theta_M)\boldsymbol{u}\frac{dv}{dy},$$ (9)

with $\Theta_M = E(v^2)$, where $E$ represents the expectation value.

We obtain the average described above as given by Toyoizumi et al [24], where also a reference activation value $v_0$ is used (note, the BCM rule works poorly for our task without this variable, see S2 Appendix):

$$\frac{d\Theta_M}{dt} = \gamma\mu(-\Theta_M + v\frac{v}{v_0}).$$ (10)

where $\gamma$ is a factor relating the time constants of the two differential equations (Eqs 9 and 10) and $\gamma$ needs to be big enough to avoid instabilities. Note that parameterizing it this way makes it easier to focus on the influence of the *ratio* between the time constants in our analyses.

For the Oja rule, we use the standard formulation from [2]:

$$\frac{d\boldsymbol{\omega}}{dt} = \mu y(\boldsymbol{u} - \alpha y\boldsymbol{\omega}),$$ (11)

where we set $\alpha = 1$. This factor leads to the asymptotic convergence of $|\boldsymbol{\omega}|^2 = 1/\alpha$ and is discussed in "Results" section.

For synaptic scaling, we use the following equation taken from [8]:

$$\frac{d\boldsymbol{\omega}}{dt} = \mu y \boldsymbol{u} + \xi(y_0 - y)\boldsymbol{\omega}^2,$$

(12)

with $\xi < \mu < 1$ and where the parameter $y_0$ determines the value at which the output is stabilizing (for concrete values see figure legends).

## Experimental settings

**Neuron with two inputs.** In the first part of the results section, we focus on the investigation of the ALL-rule on a neuron with only two inputs with varying input amplitudes and occurrence frequencies. We vary amplitudes in the interval [1, 1.5] and the occurrence frequencies using a ratio of 1 : 1 or 2 : 1. We also vary the standard deviation of input amplitude ($\sigma = 0.1$ or $\sigma = 0.2$) as well as the coincidence rates between inputs (50, 30 and 10%, measured in respect to the less frequent input in case of frequency difference). This is extended by detailed statistics how our system behaves for different annealing-rates $\rho$ and thresholds $v_a$. Finally, we compare the results from the ALL-rule to results obtained from a set of the most common learning rules under similar conditions.

**Recurrent network.** In the second part of the results section, we employed the ALL-rule for generation of *all possible* coincident combinations of $N$ external inputs (Fig 3) in a randomly connected recurrent network of $M$ neurons with sparse connectivity of $c$ connections per neuron on average. We use for connectivity a Gaussian distribution with standard deviation of $c/5$. However we are limiting this to a minimum of at least one connection onto each neuron. We also impose a limit on the maximally allowed connections, where for $c = 2$ this amounts to allowing connection numbers in the interval [1, 3]. We analyse the cases $M = 200$ and $M = 1000$, with $c = 2$ and $c = 10$ (allowed interval [1, 19]). In addition to those connections, 15% of randomly selected neurons are supplied with one connection each from randomly chosen external input neurons. We analyzed cases of $N = 3$ and $N = 5$ external inputs. Inputs can take two values: 0 or 1. For this part of the study we did not vary input amplitudes or

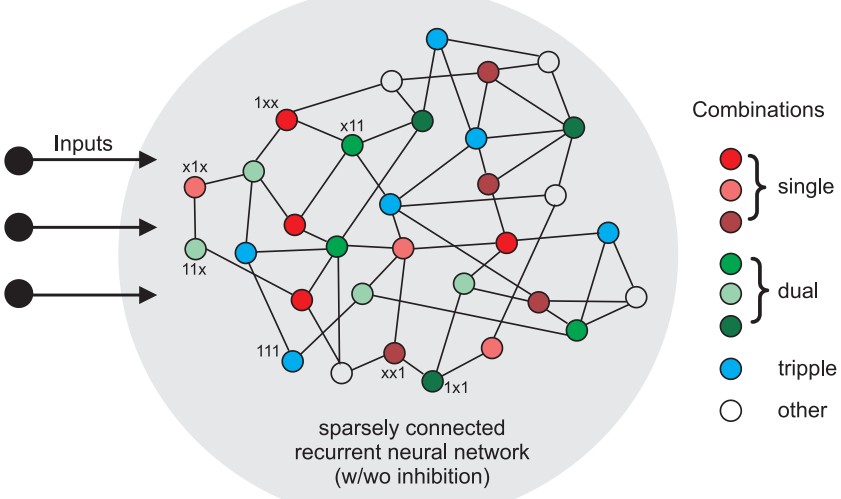

**Fig 3. Schematic diagram of the recurrent network with neurons responsive to different input combinations indicated.** "x" means input can be 0 or 1.

frequencies. The goal of this part of the study is to show that such a system can self-organize into creating output neurons that respond to *different* possible combinations of active inputs. Hence, one such neuron will then respond if a certain subset of $k$ inputs is active at the same time (AND operation) and not respond if any one of these $k$ inputs is not present. In this case the remaining $n - k$ inputs will not be able to drive this output neuron whatsoever. We were considering that a neuron is signaling for a certain input combination in case its activity is above a "classification" threshold for this combination, but below threshold for any other combination. We analyzed a set of thresholds from $v = 0.4$ to $0.8$ in steps of $0.1$.

This way, we measured how many neurons, which are signaling a possible combination, appear within a network by calculating statistics from 100 trials to generate and train a network. For this, we varied the network connection matrix trial by trial. Also, neurons in the network were generated with an annealing threshold drawn from a uniform distribution in [0.75, 0.95], which also was re-generated for each trial. Then, we present results as percentage of neurons in the network that represent a certain combination. Hence, 2% means that there were 4 neurons representing that combination in a neural network with $M = 200$ neurons and 20 neurons representing that combination in the $M = 1000$ network.

Code for the different experiments is provided in the S1 Code Repository.

## Results

First we analyze the properties of the annealing learning rules for a neuron that has only two inputs and compare those to the reference methods (BCM, Oja, Synaptic Scaling). This is started by an analytical calculation that compares annealed membrane Hebb (AMH) with annealed Linear Learning (ALL) after which we show simulation results for a wide variety of cases that cannot be captured by analytical approaches. The central finding here is that the ALL-rule allows for reliable separation between coincidence and no-coincidence cases without having to re-tune neuron parameters for different input situations. Only the BCM rule behaves similarly. However, this part is then extended by analysing more than two inputs. Here we observe now clear differences between BCM and ALL.

Finally this is followed by a study of recurrently connected networks also with more than two inputs, where we ask how reliably such a network could detect various types of coincidences. In addition to this we have performed a set of control experiments, where we added inhibition with a similar characteristic as in the cortical networks (about 20%, with constant synaptic weights and a wider convergence/divergence structure than excitation).

### Separation properties

In the following we analyze how well does the ALL-rule, as compared to the AMH-rule, separate the resulting output spike rates (coincidence case) relative to the individual rates obtained from only one input. We can here obtain analytical arguments under the assumption of independent constant inputs in the limit of few coincidences only (where the latter constraint is needed for the AMH-rule only). Then, we also complement these analytical considerations by some simulations that allow relaxing the above constraints.

Hence, we assume two constant inputs, $u_1$ and $\phi u_1$ with $\phi > 1$. For the case of the ALL-rule one can calculate weight growths over time as:

$$\begin{aligned} \omega_1(t) &= \mu_0 u_1 t + \omega_0 \\ \omega_2(t) &= \mu_0 \phi u_1 t + \omega_0 \end{aligned} \tag{13}$$

where $\mu_0$ is the learning rate before annealing and $\omega_0$ the start weight. Accordingly, the

membrane potentials are:

$$\begin{aligned} y_1(t) &= \mu_0 u_1^2 t + \omega_0 u_1 \\ y_2(t) &= \mu_0 \phi^2 u_1^2 t + \omega_0 \phi u_1. \end{aligned} \tag{14}$$

For the AMH-rule we get for the weights:

$$\begin{aligned} \omega_1(t) &= \omega_0 e^{\mu_0 u_1^2 t} \\ \omega_2(t) &= \omega_0 e^{\mu_0 \phi^2 u_1^2 t}, \end{aligned} \tag{15}$$

and the membrane potentials are given by:

$$\begin{aligned} y_1(t) &= \omega_0 u_1 e^{\mu_0 u_1^2 t} \\ y_2(t) &= \omega_0 \phi u_1 e^{\mu_0 \phi^2 u_1^2 t}. \end{aligned} \tag{16}$$

If we allow for (rare) coincidences between the two inputs then the membrane potential becomes $y_1 + y_2$ and the neuron's output will be $v_{1+2} = f_s(y_1 + y_2)$ (see Eq (2)). Due to the definitions of $y_1$ and $y_2$ the following conjecture holds: $v_{1+2} > v_2 > v_1$. As a consequence $v_{1+2}$ will eventually hit the annealing threshold $v_a$ at time $t_a$. If we now assume instantaneous annealing, then all weight growth will stop and we can ask which values will the individual outputs $v_1$ and $v_2$ have reached? This way we can assess the separation between the coincidence-driven output (which is then at $v_a$) and the other two outputs. To be able to call such a neuron an AND-operator a clear separation is needed and here we are only concerned with $v_2$, which is anyhow larger than $v_1$. Hence, we calculate for different parameters $u_1$, $\mu_0$, $\phi$ and $v_a$ how big the separation $s(t_a)$ between $v_{1+2}(t_a) = v_a$ and $v_2(t_a)$ is as $s(t_a) = v_a - v_2(t_a)$. This last step has to be calculated numerically as the resulting terms cannot any longer by analytically solved. Fig 4A shows the results. Note, that $\mu_0$ has no influence on the separation, it only determines how early/late the annealing threshold will be reached. The figure shows that only for identical amplitudes the separation between the coincidence case and the individual input case will be the same for the annealed membrane Hebb- and the annealed Linear Learning rule. For all other situations, the ALL-rule leads to a far better separation. Furthermore, note that separation is largely independent of the annealing threshold, which adds to the robustness of the annealing approach.

In panel B we show how the ALL- versus AMH-rules behave when using inputs with a Gaussian distribution in amplitudes and the same presentation frequencies for both inputs. Coincidence rate was 10%. Responses to the individual inputs are shown in orange and blue and the coincident case in green. The results are consistent with the analytics in panels A except for a slight increase in separation values due to more balanced weight growth in the simulation, because of the 10% coincidences, where the analytics could only be calculated for the limit case of 0%. The ALL-rule leads to a much stronger separation. Numbers at the bottom show the distance between the mean values of the orange and green distributions, where separability entirely ceases towards the right for AMH. Furthermore, note that the AMH-rule shows the expected exponential run-away property for the stronger (orange) distributions and the blue ones do not develop any firing rate $v$ above zero for unequal amplitudes. Using a rate-based Hebb rule (hence $g(y) = v$), would mitigate these effects as soon as the membrane potential to rate transformation approaches the Heaviside property.

## Annealed Linear Learning rule: Neuron output analysis

In the following we focus on the ALL-rule, which provides a better separation than the the AMH-rule as shown above. In Fig 5 we present histograms of neuron outputs for different input combinations. Input amplitudes are drawn from a Gaussian distribution and are

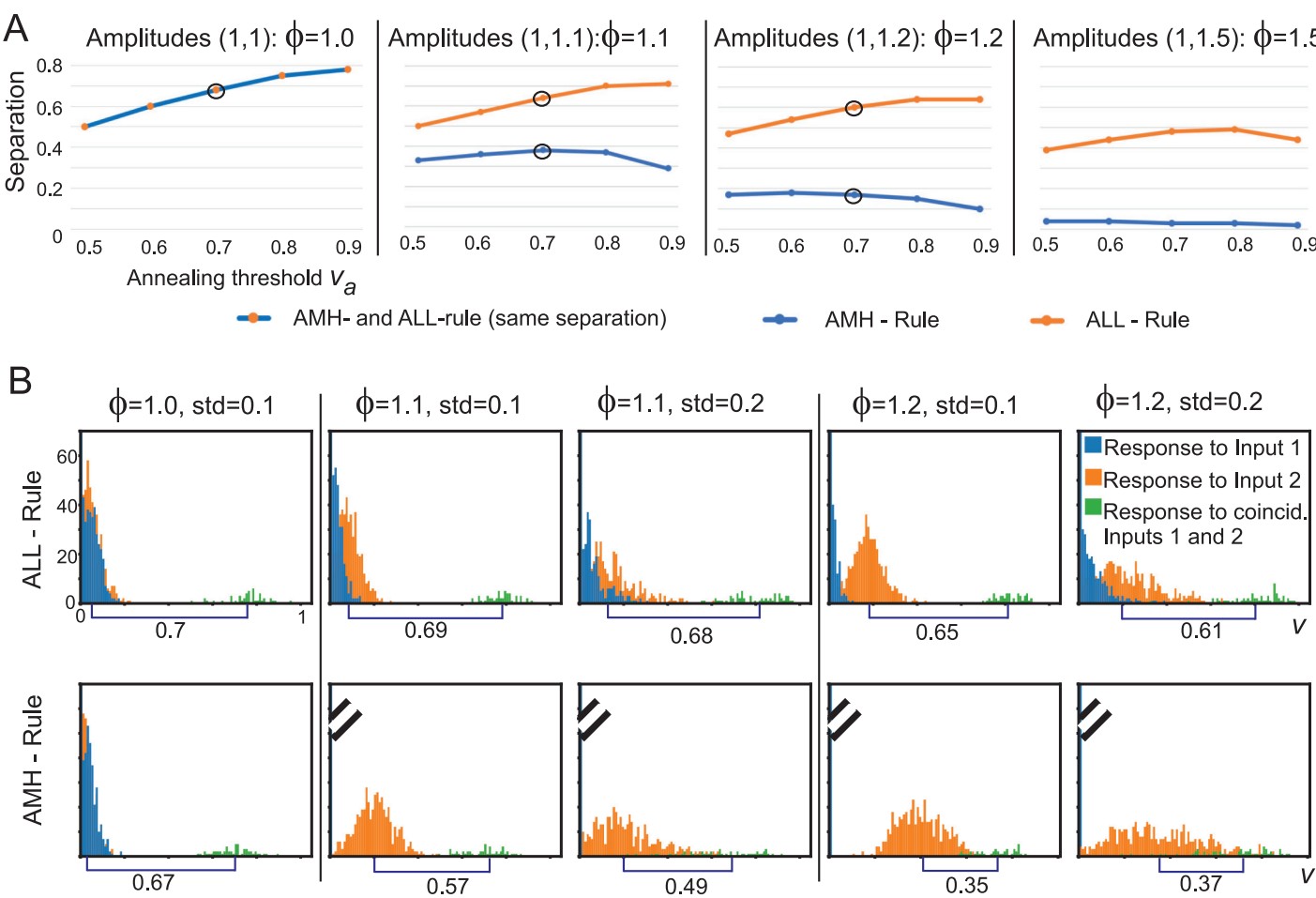

**Fig 4.** A: Separation properties calculated analytically B: Histograms of numerical results for ALL- and AMH-rules in case of Gaussian distribution of input amplitudes. In all cases, input presentation frequencies are equal. In B input coincidence is 10% everywhere; input amplitudes: mean for the blue distributions was normalized to 1.0 and for the orange ones to $\phi$; standard deviations indicated above the plots. Annealing parameters are $v_a = 0.7$, $\rho = 0.2$. Initial weights are $\boldsymbol{\omega}(0) = [0.001, 0.001]^T$ and initial learning rate $\mu_0 = 0.0005$; Euler integration with step $dt = 1$. Disks in A mark the points with the corresponding plots in B. Tilted lines are truncation marks for the blue histograms.

characterized by mean and standard deviation (see first column in Fig 5A). We use mean amplitudes of 1, 1.2 and 1.5 and a standard deviation of std = 0.1 for Fig 5. (Cases with std = 0.2, i.e., higher input variance, had been shown already in Fig 4 above).

In addition to the amplitude distribution, inputs are characterized by their presentation frequency, which could be understood as how often stimuli are delivered to the neuron by the external world. In Fig 5A, we show results in case both inputs are presented with the same frequency, while in Fig 5B results are shown in which the first stimulus is twice more frequent.

Another important input parameter is how frequently two inputs coincide at the neuron. We consider 50, 30 and 10% coincidence. When the presentation frequency of the two inputs differs, we calculate the percentage of coincidence with respect to the input with smaller presentation frequency.

In Fig 5 we show the input distributions of the neuron (left column) and the neuron output $v$ in case of coincidence in green, while the response to single inputs are blue and orange. All neuron outputs are limited to the interval [0, 1], due to the non-linear response curve (see Fig 2A).

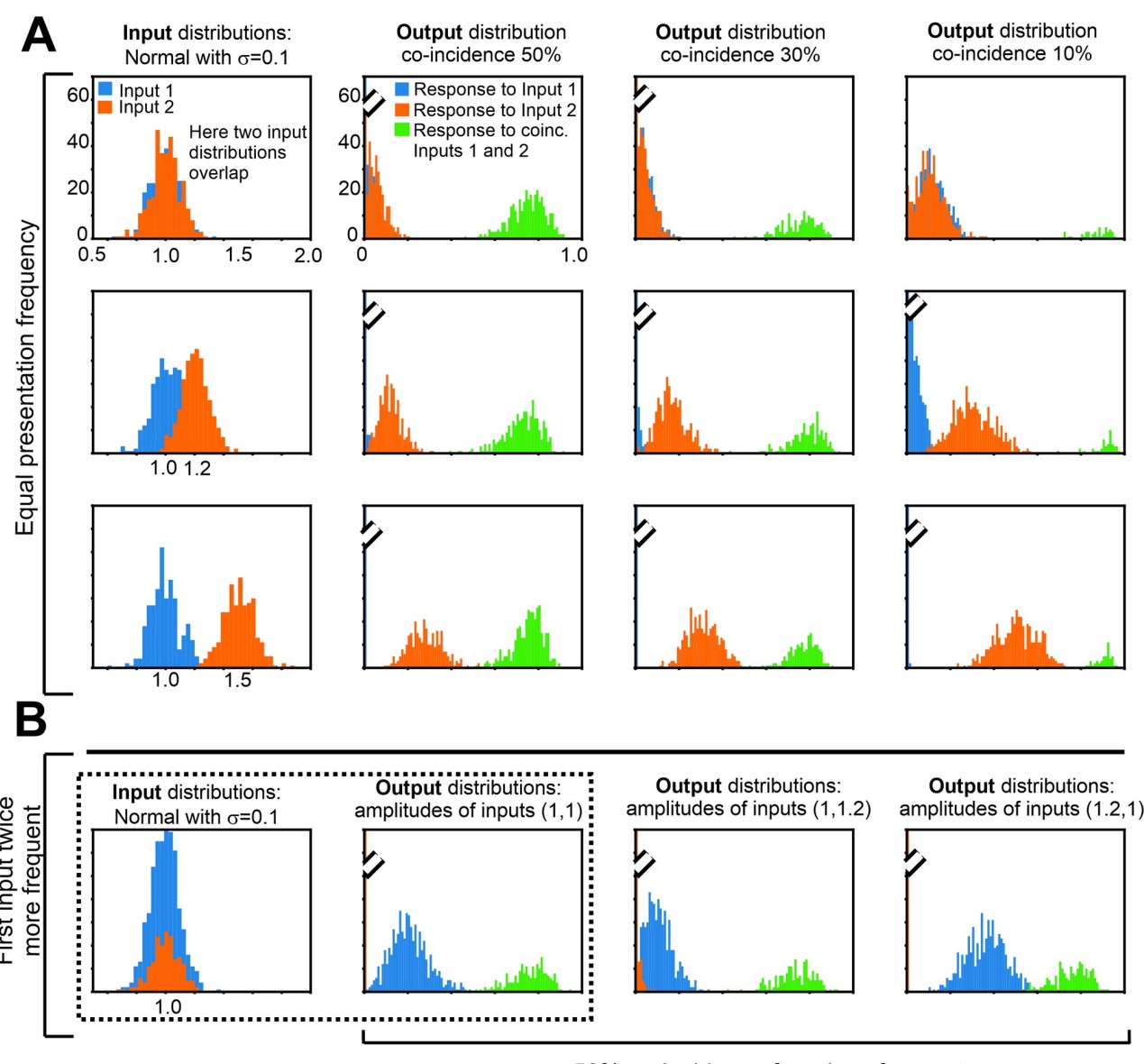

**Fig 5. Histograms of neuron inputs (first column) and outputs *v* for the ALL-rule.** A: Equal presentation frequency; B: Different presentation frequency. Parameters: $v_a = 0.7$, $\rho = 0.1$, std = 0.1. Mean amplitudes of the inputs are indicated in the first column. Initial weights are $\boldsymbol{\omega}(0) = [0.001, 0.001]^T$ and initial learning rate is $\mu_0 = 0.0005$; Euler integration with step $dt = 1$. For other parameters: see plots. Response histograms (blue or yellow) in case of amplitude or presentation frequency difference are grouping very close to zero, where we truncate the zero bin to optimize for visibility (see truncation marks).

As expected, the output for coincident inputs (green) is always the highest. We can also observe that the gap between the blue or orange histograms and the green histogram is in almost all cases quite big. Furthermore, this gap "sits at the same location" such that a unique discrimination threshold $v_d$ could be defined to differentiate coincident from non-coincident responses (e.g. $v_d = 0.6$). These properties are, thus, largely independent of input amplitudes, frequencies, and percentages of coincidence. Thus, only due to these invariances such a neuron can indeed be called "input coincidence detector" (AND operation-like). Next we will quantify the robustness of these properties.

In Fig 6 we show for the ALL-rule, how the separation of coincidence vs no coincidence varies with different annealing parameters, where we vary the annealing onset threshold and the annealing rate $v_a$ and $\rho$ (Eq (4)). We show the classification error for coincidence vs no coincidence. Classification threshold is kept at $v = 0.5$. First, in Fig 6A and 6B we present error plots in parameter space in case both inputs have the same presentation frequency and both amplitudes are equal: mean = 1, std = 0.1, with 30% (A) or 50% (B) coincidence. These are the most favorable cases from all cases shown here and one can see that the error is zero (or very small) in a very big region of the parameter space (white and light colored patch in the middle of the plots). This patch slightly decreases when amplitudes (E, F), or frequencies (C, D) of the two inputs differ, but differences between the plots remain small. Amplitude increase of the less frequent input can compensate for the frequency decrease (see G, H). The errors in the plots "above-left" the white patch are false positives, while for "bottom-right" they are false negatives.

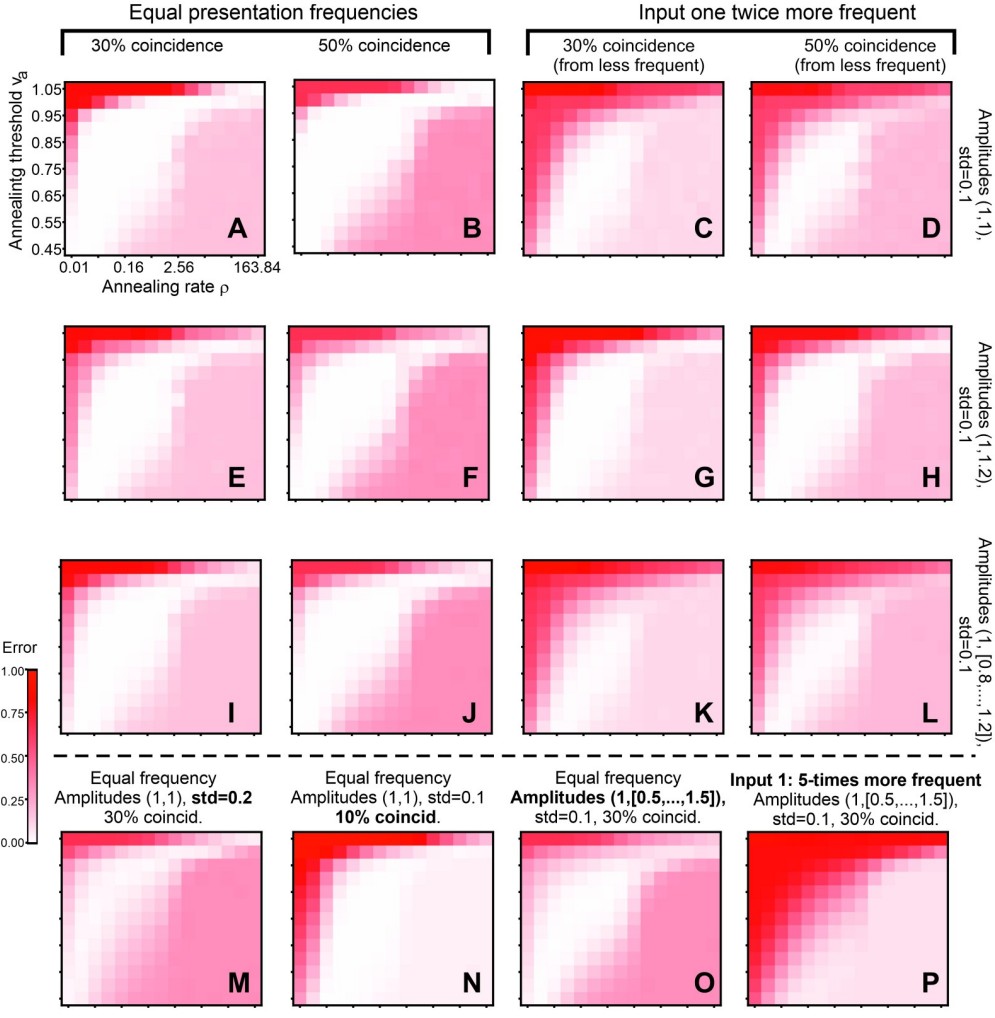

**Fig 6. Classification error (coincidence vs. not coincidence) of the ALL-rule in respect to parameter variations.** Parameters are annealing onset threshold and annealing rate. Decision threshold is 0.5. Panels (A-L) variable amplitude, coincidence and presentation frequency; panels (M-P) extreme cases: bigger variance, smaller coincidence, bigger amplitude difference, bigger frequency difference. Averages over 20 trials are shown. Initial weights are $\boldsymbol{\omega}(0) = [0.001, 0.001]^T$ and initial learning rate $\mu_0 = 0.0005$; Euler integration with step $dt = 1$.

In the third row (panels I-L) the same type of representation is shown, but for a set of amplitude differences, where the first input average amplitude is always at one, while the second input average amplitude is drawn from a set {0.8, 0.9, 1.0, 1.1 and 1.2} (uniform probability), std = 0.1 everywhere. Also in this case the error is zero in a big patch of the parameter space.

In Fig 6M–6P we present various less favorable cases to investigate the limits of the ALL-rule: higher input variance (std = 0.2, panel M), small coincidence (just 10%, panel N), wide amplitude range in the interval [0.5,1.5] (panel O), as well the case when one input is five times less frequent (30% coincidence, panel P). Except for the last, five times less frequent case, we always get a parameter region where errors are zero. In the case where one input is five times less frequent (P), however, we still get low classification errors for a large range of parameters. Note, that in this case the coincidence percentage is very small as we calculate the 30%-percentage from the less frequent input. Thus, this case is, indeed, very unfavorable.

## Comparison to reference methods

In Fig 7 we show results obtained with the three reference methods. Presented results are characteristic for the problems that these methods have with the task of input coincidence detection.

For synaptic scaling and Oja no unique separation threshold can be found and it depends on the stimulus situation. This could be resolved by using additional mechanisms (e.g. for Oja by adapting the $\alpha$ factor for each stimulus situation individually). However, case-by-case adaptation of parameters is an undesirable feature for biological systems. The Oja rule, in addition, has unfavorably overlapping distributions when input amplitudes or frequencies differ (see third and fifth columns). Note also, that Oja as well as synaptic scaling are in the existing

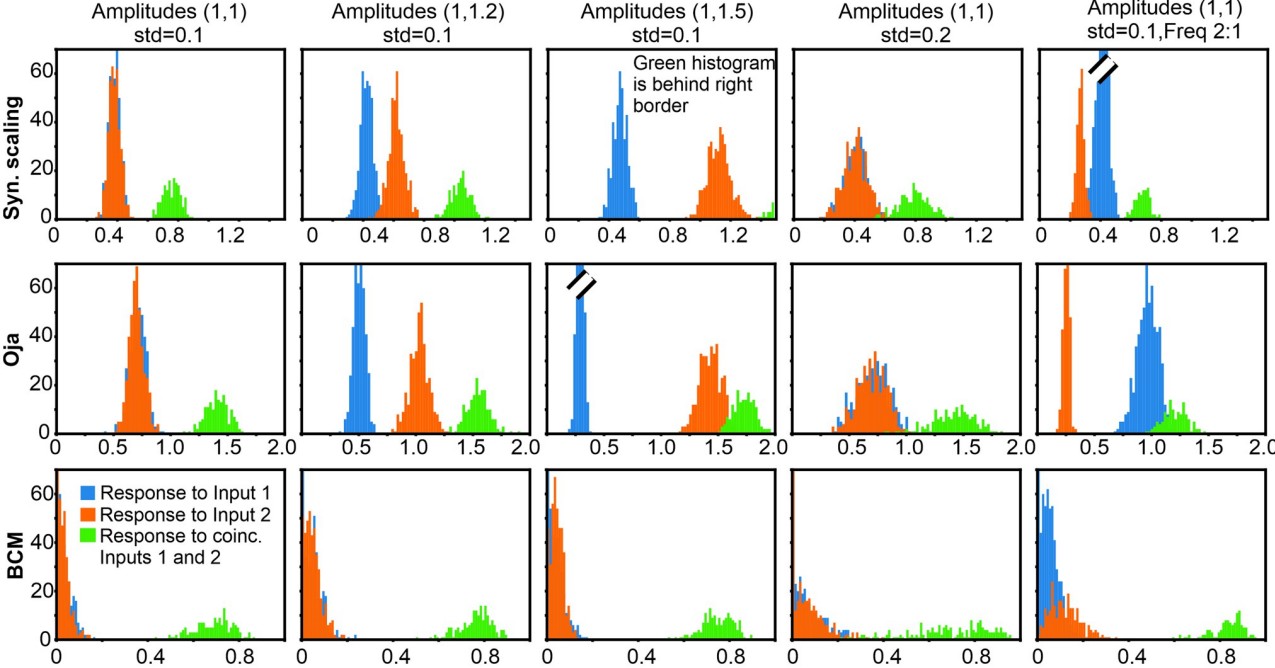

**Fig 7. Comparison to reference methods: Results for BCM, Oja and Synaptic Scaling.** Two inputs with coincidence 30% everywhere. Amplitudes and standard deviation (std) are shown above each column. Presentation frequency is equal, except in the last column where it is 2:1. Parameters: $\mu =$ 0.001. For Oja and Syn.Scaling: $\boldsymbol{\omega}(0) = [0.001, 0.001]^T$, for BCM: $\boldsymbol{\omega}(0) = [0.2, 0.2]^T$, $\Theta_M(0) = 0.2$, $\gamma = 10$ and $\nu_0 = 0.2$; Synaptic Scaling: $y_0 = -200$, $\xi = 0.01$.

literature normally used in a linear regime and cannot be satisfactorily applied after output transform (2), which we also observed (See S1 Appendix).

For the BCM rule we have investigated different variants, but we will only show the best results. In summary, when using the classical BCM-rule [3] for a 2-input system (linear case), fixed points for the synaptic weights exist, albeit one of which is always negative. Thus, this leads to unrealistic results (see S1 Appendix). This problem can be addressed by using a more advanced version of BCM introduced by Toyoizumi et al. [24]. Their formulation contains several additional parameters, which prevent negative weights. However, here distributions for single features and combinations tend to overlap and the shape and overlap of the distributions depends on those additional parameters.

Different from this, the (non-linear) version introduced by Intrator and Cooper [23] renders results which are—at a first glance—quite satisfactory and robust against input- as well as parameter variations. Thus, in all panels the same separation threshold can be used. (For this rule, however, the value $v_0$ needs to be chosen correctly, see S2 Appendix). A general observation here, though, is that the single-input distributions heavily overlay each other. Hence, different input characteristics get lost in the output. This is clearly visible when considering, for example, three inputs (Fig 8A). Here 7 different output distributions exist: 3 represent the responses for one input each, another 3 for two inputs and 1 for all three inputs. We show here three examples obtained with the BCM rule with the same parameters and same input statistics, where differences arise due to randomness in stimulus sequencing. Here always 5 distributions cluster at small activation values and 2 near an activation of 1.0. The latter consists of the 3-input case "123" and one two-input case, which is, however, not the same in the here-shown three BCM examples. The actual outcome, thus, depends on the stimulus sequences, which are randomized and, thus, different in these three examples. Different from this, the ALL-rule renders an input-output transformation which much better reflects the stimulus combinatorics, where single input responses are on the left, those for two inputs in the middle and the one that belongs to all three inputs is found on the right side of the activation axis. In Fig 8B we show, in addition the weight development for one 3-input case each for ALL and BCM, where the latter converges only after about 35,000 iterations and shows oscillations during convergence. This type of behavior of BCM for multiple inputs is generic and has also been observed by others [26]. Convergence speed can be increased by changing parameter in BCM at the cost of stronger oscillations. Note that for five inputs BCM convergence can take above 1 million iterations. Different from this, ALL converges for three inputs smoothly after about 100 iterations only and this number does not significantly increase for more inputs.

We further evaluate coincidence sorting in the three input case for the ALL and BCM rules in Fig 8C, where we investigate parameter spaces of both rules. We set two thresholds, one at 0.25 and the other at 0.75 (based on approximate boundaries of distributions in the panel A) and evaluate classification error with the assumption that responses to single inputs would remain to the left of the first threshold, two input combinations would be positioned in the middle and three input combinations would reside to the right of the second threshold. For the ALL rule, we vary parameters: annealing threshold $v_a$ and annealing rate $\rho$ for the BCM rule we vary parameters: target value $v_0$ and the time scale ratio $\gamma$; for both methods we vary steepness of the non-linear activation function by manipulating $b$ in Eq (2). Zero and small errors are visible as white-ish patches in the plots.

One can see that for the ALL rule there is an area in the parameter space with small errors and, thus, input differentiation can be obtained. Such an area is present for all three response function steepness values, though it is smaller for the very steep function. The latter is expected, as a very steep function tends to "squeeze" outputs into two classes more strongly.

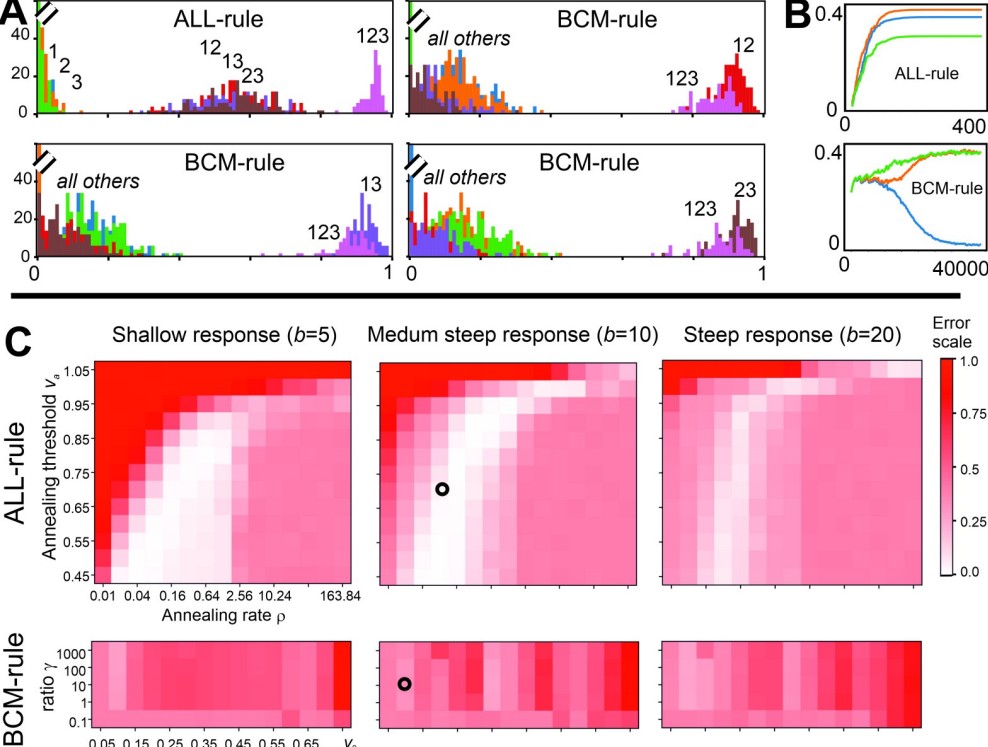

**Fig 8. Three input coincidence sorting for ALL and BCM rules.** A: Output histograms. Note that 3 examples for BCM are shown using the same intrinsic parameters but different stimulus sequencing. B: Weight development. Note the different x-axis scales. C: Parameter space analysis: Errors for classification "one active input", "two active inputs", "three active inputs" are based on response thresholds 0.25 and 0.75, averages over 20 trials are shown. Light color corresponds to good coincidence sorting. Circles in the error plots show parameter combinations for which histograms are shown in panel (A). Parameters: mean amplitude is 1 in case the input is active, $STD = 0.1$, $\boldsymbol{\omega}(0) = [0.2, 0.2, 0.2]^T$, $\mu = 0.001$, pair-coincidence 30% for every possible combination (12, 13, 23) in respect to that pair, triple coincidence for 123: 6%; for BCM: $\Theta_M(0) = 0.1$; Euler integration with $dt = 1$ in all cases.

By contrast, for the BCM no parameter combination brings small classification errors. Also, a different steepness of response function does not mend the situation. As already shown in Panel A, BCM tends to divide outputs into two extremes, thus no combination sorting property in case of more than two inputs is obtained by BCM.

In Fig 9 we further analyze ALL rule in case of amplitude variations. We investigate cases where mean amplitudes for the three inputs are (1.0, 1.0, 1.2), (1.0, 1.0, 1.5) and (1.0, 1.2, 1.5). One again can see substantial areas (light color) in parameter space where correct input sorting is happening (see top row in Panel A). At the bottom of Panel A we show two output histograms for instances marked by two circles in the parameter space above. In those instances outputs can be differentiated between "one active input", "two active inputs" and "three active inputs" given chosen thresholds with a small error.

Finally, we investigate five input cases for ALL and BCM (Intrator-Cooper) rules. In Fig 9B we show outputs after learning obtained for different input combinations (there are 31 possible combinations with at least one active input for five inputs). In this case all inputs have a value of 1 (no amplitude variation) and in the learning phase all input subsets are provided with equal probability. For the ALL rule we reliably and consistently obtain outputs sorted by the number of active inputs, as shown in the left plot in Fig 9B. By contrast, for BCM the situation

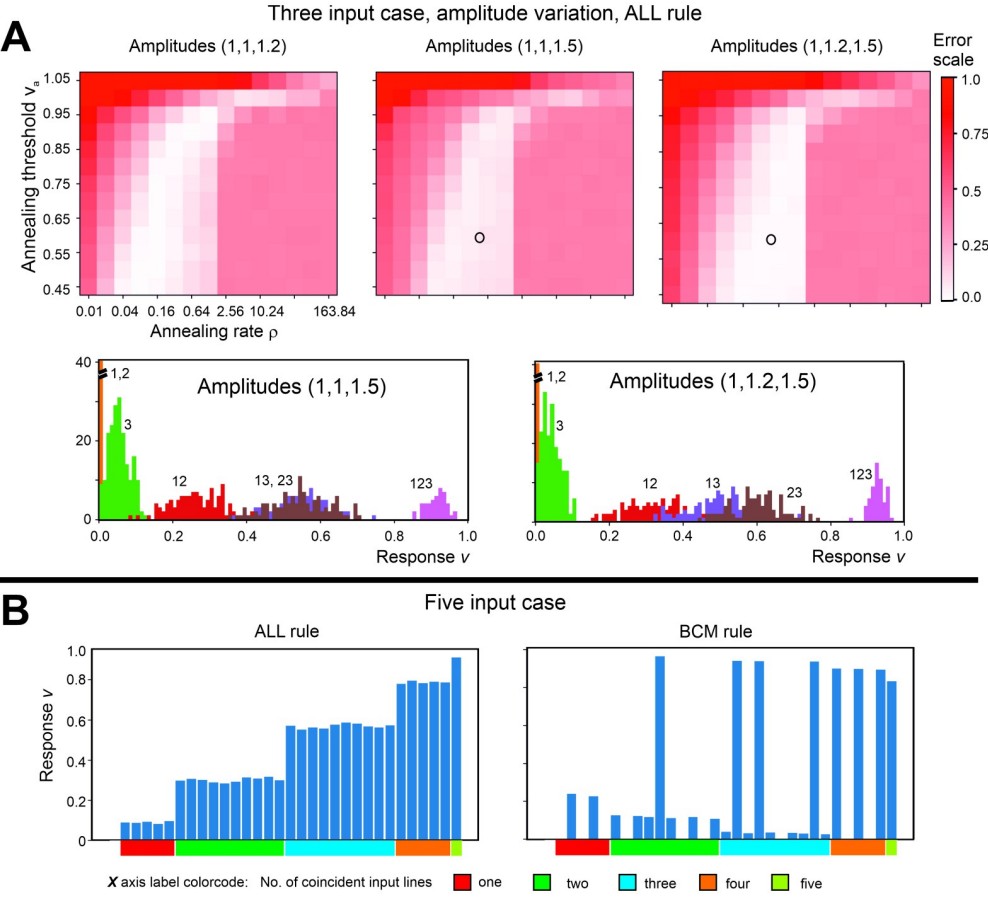

**Fig 9. Input coincidence sorting properties under more variable conditions.** A: Results for the ALL rule for the three input case with amplitude variation. Errors for classification "one active input", "two active inputs", "three active inputs" are based on thresholds 0.25 and 0.75. Light color corresponds to good coincidence sorting. Parameters: average amplitude provided above the plots, $STD = 0.1$, $\omega(0) = [0.2, 0.2, 0.2]^T$, $\mu(0) = 0.001$, pair-coincidence 30% for every possible combination (12, 13, 23) in respect to that pair, triple co-incidence for 123: 6%; plots show averages over 20 trials. Histograms of individual runs below correspond to the two circles in parameter plots above. B: Results on input coincidence sorting for ALL and BCM (Intrator-Cooper) rule for a five input case. For 5 inputs there are 31 possible combinations of neurons driven by $n \geq 1$ inputs: $5 \times 1$, $10 \times 2$, $10 \times 3$, $5 \times 4$ and $1 \times 5$ inputs as indicated beneath the abscissa. Parameters: $\omega(0) = [0.1, 0.1, 0.1, 0.1, 0.1]^T$ $\mu = 0.001$, binary subsets of five presented in equal probability, random order, Euler integration with $dt = 1$ in both cases, for ALL: $\nu_a = 0.7$, $\rho = 0.1$, for BCM: $\Theta_M(0) = 0.2$, $\nu_0 = 0.4$, $\gamma = 10$.

is different ás can be seen on the right in Fig 9B. Note, where there is no blue column, the output is zero. Outputs by BCM are essentially sorted into two classes, close to zero and close to one, similar to the result shown in Fig 8A. Also in a similar manner, the outcome is variable and depends heavily on the actual input sequence. Thus, the BCM rule cannot sort five input coincidences.

Hence, ALL-rule has unique properties in respect to other rules in coincidence sorting or detection.

## Recurrent networks with the ALL-rule

First we demonstrate that we can obtain cells representing all possible input combinations in a recurrent network. In Fig 10 we provide a box plot for the number of different combinations

obtained for $N = 3$ or $N = 5$ external inputs in case of $M = 200$ neurons in a network. Statistics are shown for 100 randomly generated networks. For this we count, after learning, how many neurons respond, for example, to an input combination of "x11xx". Such a neuron, shown with green index "12" (decimal for the binary code 01100) in the panel B, thus, requires inputs 2 and 3 (encoded as "1") to be active, where the other inputs may or may not be present (encoded as "x"), but they will not be able to drive this neuron on their own. One can see that for $N = 3$ the number of cells representing different combinations is essentially uniformly distributed, while for $N = 5$ the number of neurons representing single inputs is higher than the rest. As expected standard deviations are high but, in spite of this, for any of the possible combinations there are always at least a few cells that represent them.

It is here important to note that this network does not produce an excess of neurons that respond to the condition "other" (about 7 aut of 200 cells do this in the 5-input case, panel B). "Other" means that a neuron would code, for example, for "x1x1x" as well as for "11xxx" and possibly for even more different combinations. If self-organization were driven by a pure random process a very strong excess of such neurons would be expected, which is not the case here. Hence, our networks, indeed, self-organize into a set of input-combination selective neurons.

In Fig 11 we analyze how the proportion of different combinations change with varying decision threshold (A,C) and for the same decision threshold but in different network architectures (B,D). We quantify how many neurons are—on average—selective for *any* input combination. To achieve this, we first sum the number of neurons that represent the same *type* of input combination: e.g. single input. Then we divide this sum by the number of possible type-identical combinations. For example, for the $N = 5$-case there are 5 single, 10 double, 10 triple, 5 quadruple and 1 quintuple possible combinations existing. Hence, percentage plots in Fig 11

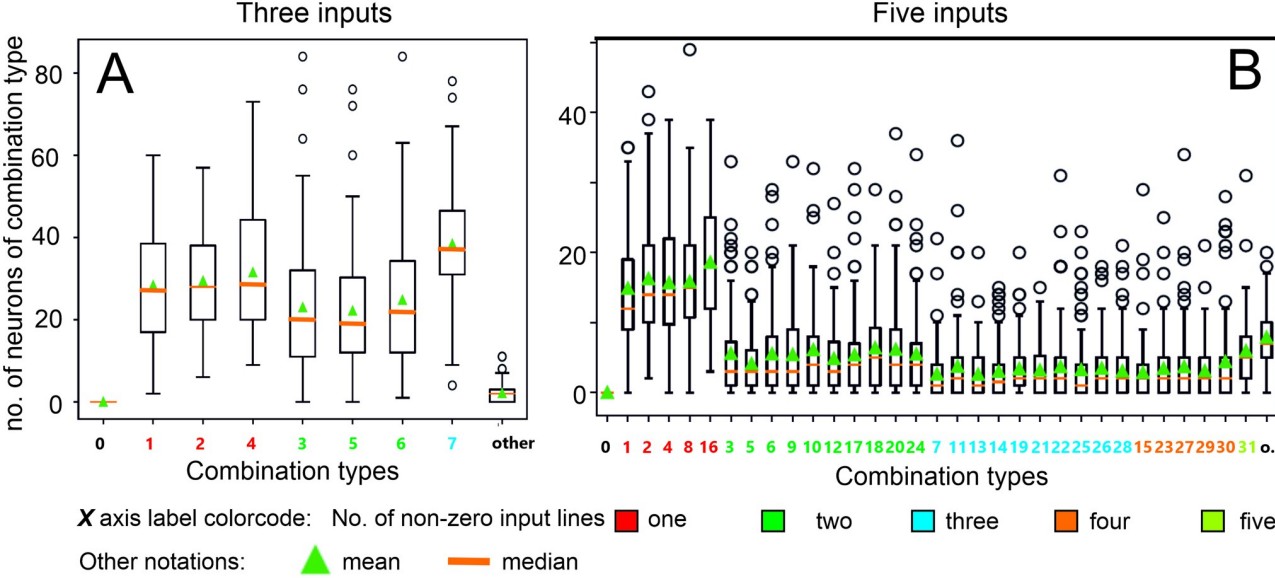

**Fig 10. Box plots for the number of neurons representing different combinations for the ALL-rule.** A: Input number $N = 3$. B: Input number $N = 5$. Combinations are aligned in ascending order of active inputs, with color code indicating the number of inputs, see legend at the bottom. Combinations are indicated by decimal numbers corresponding to binary set notation (e.g. "3" means the combination: 00011, where only the two last inputs are active). "o" means other, where this denotes occurrences of cells signaling several different combinations. The size of the neural network is $M = 200$, average connectivity $c = 2$, annealing parameters are: annealing rate $\rho = 0.3$, where the annealing threshold $\nu_a$ for each neuron individually is drawn from a uniform distribution [0.75,0.95]. Decision threshold is 0.7. Initial weights are chosen from Gaussian distribution with mean = 0.001 and std = 0.0002. Initial learning rate $\mu(0) = 0.0005$. Euler integration with $dt = 1$. Median, mean and standard deviation are shown on the basis of 100 trials.

do not sum up to 100. However, to also be able to show the strong difference between combination-selective versus non-selective ("other" + "sub-threshold" + "sustained") neurons, we provide the total percentage of the combination-selective neurons, too (numbers in italics at the top of each plot). Standard deviations are of the same order of magnitude, as shown in the box-plot above and omitted here to make the diagrams better readable.

Decision threshold dependence is analyzed in parts A and C for $N = 3$ and $N = 5$, respectively. All combinations are represented with an increasing prevalence of more-complex combinations for higher threshold values. Single cases are over-represented for smaller thresholds, where this over-representation decreases with increasing decision threshold. However, the qualitative outcome remains the same, that *different* combinations exist in a network, irrespective of threshold value.

While all combinations are present, architectures with a bigger number of inputs (200–10, 1000–10) favor higher-order combinations of inputs (see prevalence of 3-combinations in Fig 11B for architecture 200–10 and 1000–10 and prevalence of 5-combinations in Fig 11D, appropriate architectures).

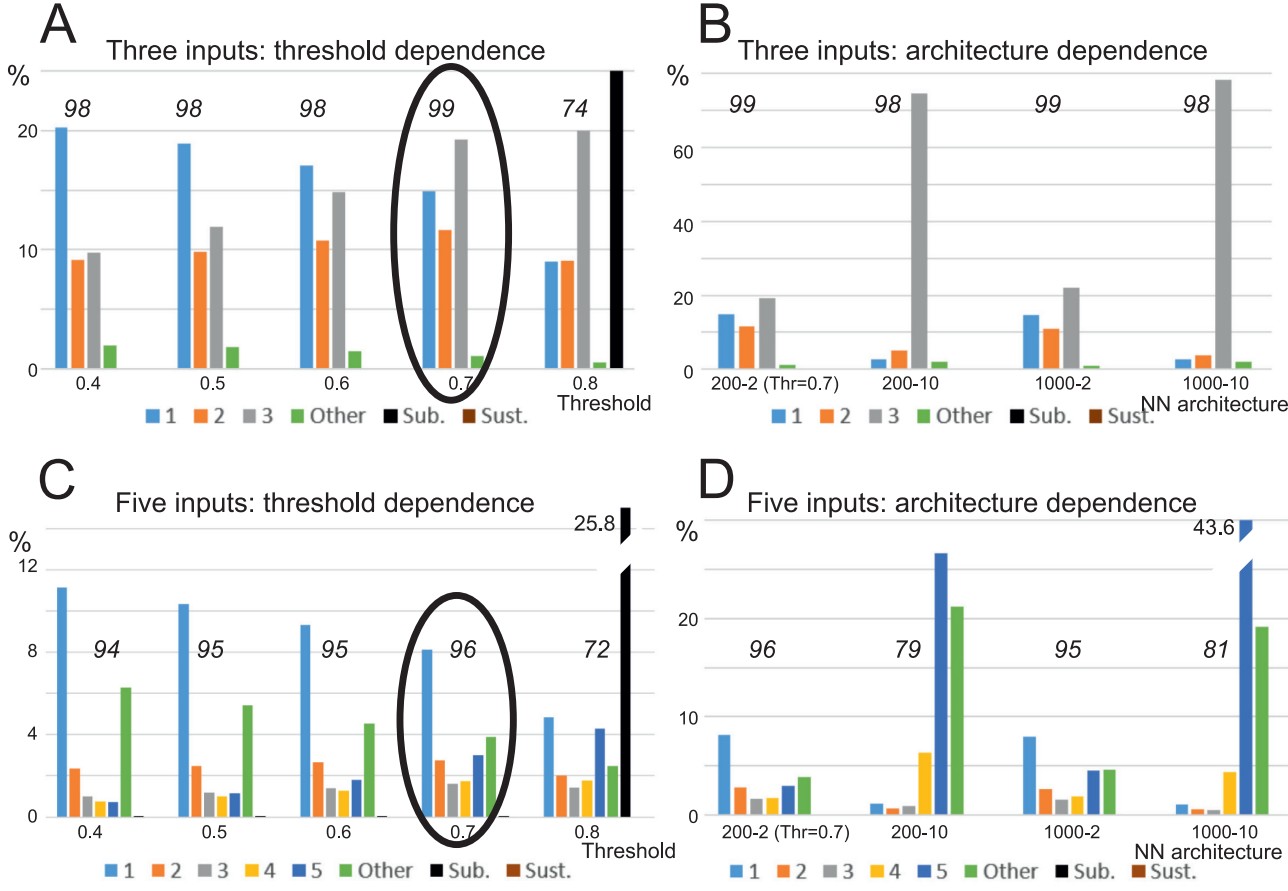

**Fig 11. Different combination distribution based on decision threshold and neural network architecture.** A and B: three input case; C and D: five input case. Numbers 1 to 5 indicate combinations responsive to corresponding number of inputs; "Other" represent cells signaling more than one combination (see text for explanation), "Sub." denotes sub-threshold cases, while "Sust." denotes sustained activity, which does not subside after switching off the inputs, which does not happen here (but in the baseline, see Fig 12). Neural network (NN) architecture notation: "No of neurons"-"connectivity" (*M-c*). Numbers above column groups denote percentage of combination-selective neurons (vs. "Other" and "Sub." neurons). Initial settings and learning parameters as in Fig 10. Note, Fig 10 corresponds to the results indicated by ovals.

The green columns in the histograms show "Other" cases, which count all the cells in the network that are active with two or more combinations, where one is not the subset of the other. This number is not substantial when connectivity is low $c = 2$ and higher if we use connectivity $c = 10$ in five input case (see panel D). However, note that also in this case there are still around 80% of combination specific cells existing and only 20% others. If the decision threshold becomes too high (see for the threshold value 0.8 in A and C), sub-threshold cases emerge (black column). These are cases where the neuron may "fire" but never reaches decision threshold. None of the networks that we trained this way showed sustained activity, which is a type of activity that persists after the inputs have been switched off (but see next).

Results can be compared to baseline performance, where the weights obtained by the ALL-rule are randomly reshuffled (permuted) in between connections in the network, while the general network connectivity pattern (which neuron connects to which other neuron) remains the same. The percentage of different cases is shown in Fig 12, column group "permuted". Here we see that both, for $N = 3$ (A) and $N = 5$ (B), we only have a few percent of neurons responding to single inputs only, whereas essentially no more-complex combinations emerge (compare to columns "learned" plotted on the left). Instead, for baseline most of the cells remain sub-threshold. The question naturally arises whether this is just a scaling effect? Hence, to investigate if we can get more useful above-threshold combinations with bigger weights, we increased all weights in the baseline by 1.5 or by 2 (columns "permuted x 1.5" and "permuted x 2"). Here we get a few more single responses and, as discussed above, also more "Other" responses (numbers in green), but now also sustained activity emerges (brown column in the diagrams) and dominates for "permuted x 2". Hence, the network activity does not come to rest after stimuli have been removed. Thus, this baseline shows that the ALL-method, suggested in this study, allows generating in an unsupervised manner neurons selective for *specific* combinations of inputs (low number of random="other" combinations) in a stable way, hence, without leading to sustained activation.

## Network with inhibition

This study focuses on the stabilizing effects of annealing in excitatory networks, which otherwise would be prone to effects like sustained activity as shown in the baseline study above.

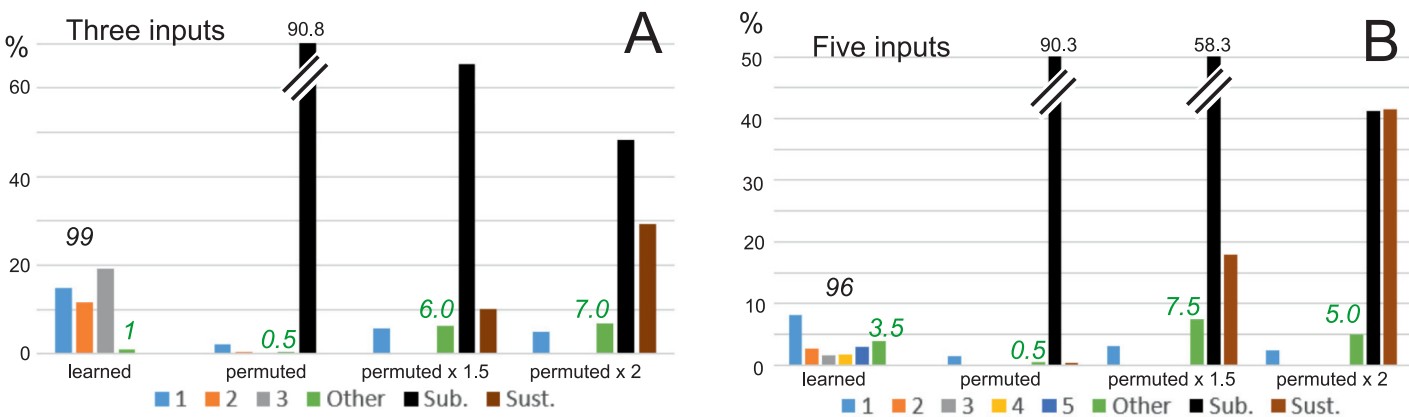

**Fig 12. Comparison to baseline.** A: Three input case, B: Five input case. The column group "learned" shows performance of the ALL-rule, $M = 200$, $c = 2$; copied from Fig 11; "permuted" is for the case with learned weights randomly permuted; "permuted x 1.5" and "permuted x 2" for cases with permuted weights multiplied by 1.5 and 2, respectively. Decision threshold kept at 0.7 everywhere. Abbreviations: "Sub." = sub-threshold, "Sust." = sustained activity. Green numbers denote percentage of "Other".

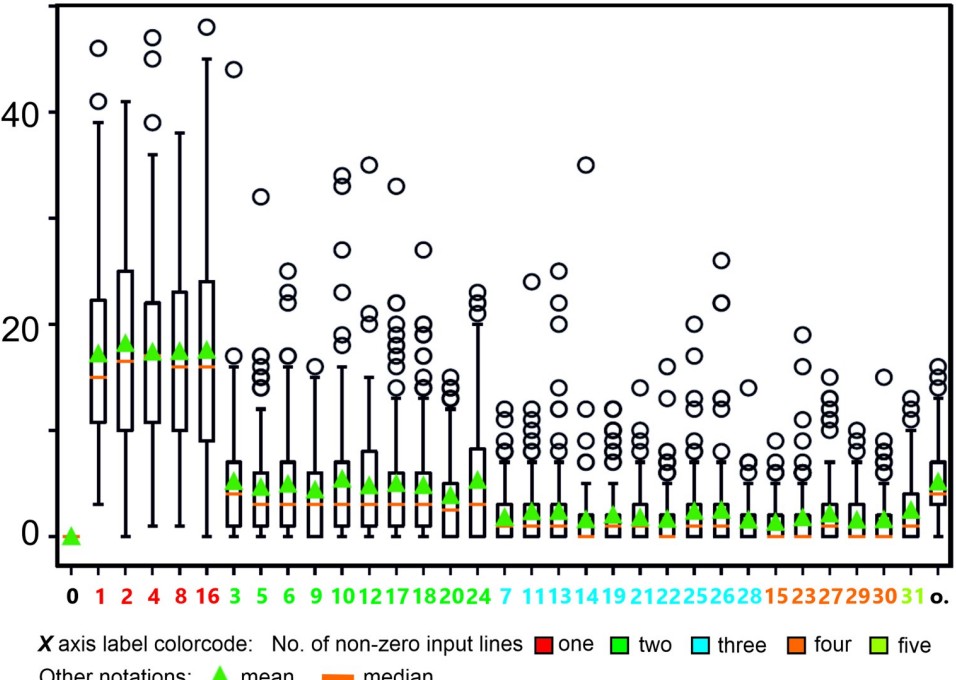

**Fig 13. Box plot for different combinations of inputs for the cases $N$ = 5 inputs for the ALL-rule with 20% inhibitory cells.** Combinations are aligned in ascending order of active inputs, with color code indicating the number of inputs, see legend at the bottom. Combinations are indicated by decimal numbers corresponding to binary set notation (e.g. "3" means the combination: 00011, where only the two last inputs are active). "o" means other, where this denotes occurrences of cells signaling several different combinations. The size of the neural network is $M$ = 200 (excitatory cells) with 40 inhibitory cells added. Average connectivity of excitatory cells onto excitatory cells is $c$ = 2; connectivity onto inhibitory cells $c$ = 20. Each excitatory cell, in addition, is given 10 inhibitory connections, with a fixed weights of 0.01. Annealing parameters are: annealing rate $\rho$ = 0.3, where the annealing threshold $v_a$ for each neuron individually is drawn from a uniform distribution [0.75,0.95]. Decision threshold is 0.7. Initial weights for excitatory inputs are chosen from Gaussian distribution with mean = 0.001 and std = 0.0002. Initial learning rate $\mu(0)$ = 0.0005. Euler integration with $dt$ = 1. Median, mean and standard deviation are shown on the basis of 100 trials.

Intuitively, inhibition should not interfere (rather help) with stabilization, but will it affect responses to the input combinatorics?

In Fig 13 we show a boxplot for the number of cells signaling different input combinations in a network with 200 excitatory cells and 40 inhibitory cells (20%). We use a connectivity of the excitatory network $c$ = 2, and $N$ = 5 inputs. In addition, each excitatory cell receives input from ten randomly chosen inhibitory cells. Inhibitory connections are not trainable and their weights are set to 0.01 each. This leads to a total inhibitory strength converging at any target cell, which is only moderately smaller than the learned excitatory weights for which we calculated that we get an average total excitatory weight of $\sim 0.5$. Each inhibitory cell receives inputs from 20 randomly chosen excitatory cells in the network. The plot in Fig 13, shown here, is similar to the one obtained without inhibition (see Fig 10B), where the numbers of cells responsible for combinations are only slightly lower in the case of inhibition. This shows that realistic inhibition, added to the network, does not fundamentally change the behavior of such networks.

## Discussion

In this study we have introduced an unsupervised synaptic plasticity rule with learning rate annealing that leads to weight stabilization and useful output sorting in case of different input

coincidences even for different input amplitudes and occurrence frequencies. To achieve this we have made two modifications of the traditional Hebb rule:

- We reduced the influence of the neuronal output onto learning to an all-or-none behavior by using the Heaviside function with a threshold $\eta \geq 0$. This way learning starts as soon as the neuronal activity is larger than this threshold but does not depend on the actual magnitude of the neuronal activity.

- We used annealing of the learning rate, as soon as the neuron has reached a certain output level. While annealing is a well-known supplementary technique in many, also unsupervised, approaches [15–18] we use it as the *main* mechanism to stabilize learning.

We have shown that with the ALL-rule neurons can learn coincident feature detection, similar to an AND operator (Fig 5). When restricted to two inputs from the here analyzed other rules it is only the BCM rule that achieves this reliably, too. However, we found that BCM cannot sort more than two inputs. This is due to its intrinsic multiple non-linearities. While weights will converge, the actual location of the BCM fixed points can not be predicted in the general case, and neurons may respond too strongly to few inputs and too weakly to combinations of more (see Figs 8A and 9B). The ALL rule, on the other hand, can solve the sorting problem, too.

In addition, we found that different neurons in a network become specific for different feature combinations when using the ALL rule. Remarkably, this happens reliably even in networks that exclude balancing effects due to inhibition and we have shown that such excitatory networks do not run into a regime of uncontrolled sustained activity. Somewhat expected, when adding realistic inhibition to the network findings remain similar and the here-observed characteristics only change for overly strong point-wise acting inhibition, which appears unrealistic when considering cortical networks.

## Biophysics

**All-or-non learning.** The use of the Heaviside function for Hebbian learning (Eq 8) provides, from a theoretical perspective, several clear advantages because it leads only to linear weight growth. Different from this, the membrane Hebb rule, which uses the membrane potential to drive learning (Eq 6), leads to exponential weight growth and a strong run-away effect of the weights that belong to the stronger inputs (see Fig 4B). Furthermore, (especially at dendritic spines) it appears that the post-synaptic depolarization effects, that influence $Ca^{++}$ influx through NMDA channels, which determine LTP, have an all-or-none effect on plasticity. The absolute values for $Ca^{++}$ within the dendrite required for the induction of synaptic plasticity have been estimated as 150–500 $nM$ for LTD and >500 $nM$ for LTP [27]. Furthermore, it has been measured that a single EPSP can raise the $Ca^{++}$-level to 700 $nM$, where a pairing of post-synaptic depolarization with synaptic stimulation would even drive it up to as much as 12 $\mu M$ [28].

Based on these findings [29] had designed a model of plasticity in spines that predicts that an EPSP resulting from the activation of a single synapse is sufficient to cause a significant $Ca^{++}$ influx through NMDA receptors. This is in line with experimental data [30–32]. As a consequence, it appears that every post-synaptic back-propagating spike or dendritic spike will be enough to lead to substantial $Ca^{++}$ influx to trigger plasticity (at a spine). This argues for a sharp transition of the post-synaptic learning influence, where the use of the Heaviside function would represent a limit case. Sigmoidal transition functions similar to Eq (2) could be used instead, where results for this study will be little affected if the sigmoid is steep enough.

**Learning Rate Annealing.**    In 1998, Bi and Poo [33] had shown that the change in EPSP amplitude is inversely related to the size of the EPSP when employing a plasticity protocol. Hence, large synapses grow less than small synapses. This is potentially a ceiling (saturation) effect of LTP and could, in theoretical terms, indeed be captured by a learning rate annealing mechanism. This, however, points to a core problem: For theoreticians the learning rate is just a single variable and learning rate annealing is essentially just an abstraction of meta-plasticity. Linking this to complex multi-faceted biophysical processes, thus, remains difficult. There is a wealth of literature that suggests that the reduction of LTP, due to meta-plasticity, could rely on effects that influence NMDA receptors [34–38]. However, the time course of this might be too fast as these effects seem to decay within about one hour [34]. Stimulus driven annealing ought to be able to act rather on longer time-scales because the animal may only now and then encounter the relevant stimuli. Longer lasting reduction of LTP could be obtained by mechanisms that operate on its later phases (late-LTP) [39–42] suspected to be essential for establishing synaptic consolidation. However, any potential role of this mechanisms in meta-plasticity related to annealing effects remains unknown. Nevertheless, it seems conceivable that neurons reduce their 'learning-efforts' by reducing the synthesis of some relevant biochemical components using a saturation-driven kinetics, as soon as the neuron's activity has grown enough, which could be understood as learning rate annealing.

**Summary.**    The above discussion provides evidence that the here-assumed novel mechanisms of all-or-non learning paired with annealing are compatible with the biophysics of synapses (especially when considering spines). It is furthermore noteworthy that the biophysical "machinery" to implement the ALL-rule is relatively simple, which is different for most other advanced unsupervised rules (see next).

## Other unsupervised rules

**BCM.**    In the theoretical literature, learning rate annealing is a very widely used mechanism applied with different learning rules and for different purposes [17, 18, 43, 44]. Notably, the BCM rule also has a mechanism built in that could be understood as annealing. Its threshold $\theta$ relies on the time-averaged level of postsynaptic firing. Thus, if firing levels are maintained at a high level, this threshold shifts, making LTP harder to obtain. With some tuning, this rule was also able to solve the simple AND-operator task investigated in this study. The weight development of BCM, however, does not reflect the ordering of input coincidences in any reliable way (see Figs 8 and 9) and the location of the different output distributions in activation space cannot be predicted for neurons with several inputs. An additional undesired aspect of BCM is that convergence can be very slow for multiple inputs. This had been observed in a recent study [26] and we also found that for five inputs sometimes above one million iterations were needed until convergence. Increasing the learning rate does not much help here as this way quite strong weight oscillations can occur. When considering that we are here dealing with the presentation of external stimuli that "come from the world", it is impossible for an animal to learn feature constellations using BCM due to delayed convergence. By contrast, the ALL rule converges (by construction always without oscillations) in about 100 iterations, where this number is not much affected by the number of inputs.

Given the complexity of more advanced versions of BCM (e.g. Intrator-Cooper), it is also unclear how this could be modeled in biophysical terms. In particular, also in view of the fact that saturation-driven kinetic mechanisms, which operate on one or more compounds needed for LTP, do not map well to this rule.

**Synaptic scaling.**    Synaptic scaling has been suggested as a possible mechanism to achieve targeted weight-growth, too, and scaling operates on rather long time scales, slower than

learning. Hence, one aspect concerns the question to what degree the ALL-rule might relate to synaptic scaling [5]. Scaling assumes that neurons "want" to achieve a certain target activity [7] and that synaptic changes are driven by this target. Hence, this is indeed related to the operation of the ALL-rule. Alas, the existing mathematical formulations where (Hebbian) plasticity is combined with some scaling term [8], do not reliably lead to this property. Different from this, the ALL-rule does achieve this in a robust manner, where—for the purpose of this study —we have set the target activity to relatively high values, which allows getting the AND-operator property. However, due to the design of the annealing mechanism, other target values can also be obtained by using a different (lower) annealing threshold $v_a$.

**Oja's rule.** This rule did not allow us to obtain in any reliable way the simple AND-operator property and had—as a consequence—not been further investigated.

**Summary.** The central problem of the above discussed learning rules appears to be that, while they all converge, the locations of the weights' fix points are not directly coupled to the (average) stimulus intensity given approximately by the product of amplitude with occurrence frequency of the stimuli. Even in cases where the stimulus statistics are identical, different stimulus sequencing will drive weight development into different fix points on their attractor landscape. This is different for the ALL rule, which leads to a rigorous "sorting" of the outputs according to their driving stimuli even for multiple inputs.

## Comparing to other learning principles

Clearly, reinforcement learning and supervised learning would be able to achieve the tasks investigated in this study, too. Both, however, require evaluative feedback in the form of rewards or by use of an error function. While evaluative feedback can help achieving more discriminative results in learning tasks, not excluding the here-addressed task of coincidence sorting, the origin of error terms in biological systems is a large unresolved question in its own right [45]. We had discussed that there are formal similarities between our rule (Eq 8) and Rosenblatt's perceptron [22], but for the perceptron an error term is needed. Note, that error terms in biological systems do not come "for free". Any system using evaluative feedback needs additional components and complex processes. The multifaceted properties of the dopaminergic system in the animal brain (i.e. a reward-processing system which, however, also strongly reacts to just a novelty signal) testifies to this complexity [46, 47]. Different from this, our method is non-evaluative and performs a process of self-organized stimulus sorting in a single neuron. Any potential ecologically meaningful evaluation could then come on top and, for example, reinforcement learning of a beneficial behavioral policy could make use of the responses of our feature-combination specific neurons.

## Limitations

This study has focused on a stationary environment, where the statistics of the inputs does not change between training and testing of the system. It is however, straightforward to complement this with a decay term (forgetting) of the weights with which the system can recover its learning rate. Thus, given the fast convergence properties of the ALL-rule, changes in the environment, which happen usually on a slow time-scale, could be accommodated this way.

Currently the ALL-rule leads only to weight growth. Forgetting would be a passive, possibly slow, mechanisms to reduce weights. Different from this, active weight reduction can also be achieved with a mechanism for long term depression (LTD). This can, for example, be done by using a sigmoidal function $G(y - \eta)$, $\eta > 0$ with values between $-1$ and $+1$ (or the Sign function) instead of the Heaviside function, which will lead to weight reduction for $y - \eta < 0$. We

are currently investigating both aspects (forgetting as well as LTD), but this goes beyond the scope of the current study.

Furthermore, we found that the ALL-rule is quite robust to variable stimulus occurrence frequencies and variable amplitudes. Only large amplitude differences will indeed harm performance. However, there is strong evidence that input normalization is a powerful mechanism in many different brain areas ([19], and see review [48]). Note that a factor of 1.5—like for the amplitudes in our experiments—is clearly within the normalization regimes for many of these experimental findings provided in the aforementioned studies [19, 48]. In an ecological setting animals have no control over how often a stimulus will occur and robustness against this kind of variability is useful for the learning process as also observed with the ALL-rule. In addition to this, normalization mechanisms can be used to ameliorate negative effects of amplitude variations.

The here investigated networks are small but their general connectivity pattern appears realistic relative to the here-used neuron numbers. Furthermore, similar types of networks have been used in many studies that address the problem of reservoir computing [49, 50]. The focus on excitation had been chosen to demonstrate that even such networks will stabilize but some general inhibitory connectivity had been introduced, too. More targeted inhibitory connections (e.g. lateral inhibition) will begin to make sense only as soon as some topology is introduced in such networks.

## Conclusions

With the mechanisms employed here we demonstrated that neurons can learn to respond to specific input combinations in an unsupervised manner. This can only be achieved if the system reacts in a rather invariant way to stimuli of different amplitude and occurrence frequency, which is assured by annealing. We believe that this specificity for input combinations may be of ecological relevance for an animal, because it allows learning to respond to sets of inputs that might indicate situations with different—positive or negative—valance, where—on the other hand—individual features might be irrelevant.

## Supporting information

**S1 Appendix. Additional analyzes of reference methods.**
(PDF)

**S2 Appendix. Parameter analysis for the BCM rule in case of two inputs.**
(PDF)

**S1 Code Repository. Code for obtaining result figures presented in this manuscript.**
(ZIP)

## Author Contributions

**Conceptualization:** Minija Tamosiunaite, Christian Tetzlaff, Florentin Wörgötter.

**Data curation:** Minija Tamosiunaite.

**Formal analysis:** Minija Tamosiunaite, Florentin Wörgötter.

**Funding acquisition:** Christian Tetzlaff, Florentin Wörgötter.

**Investigation:** Minija Tamosiunaite, Florentin Wörgötter.

**Methodology:** Minija Tamosiunaite, Christian Tetzlaff, Florentin Wörgötter.

**Project administration:** Florentin Wörgötter.

**Resources:** Florentin Wörgötter.

**Software:** Minija Tamosiunaite.

**Supervision:** Florentin Wörgötter.

**Validation:** Minija Tamosiunaite, Christian Tetzlaff.

**Visualization:** Minija Tamosiunaite, Florentin Wörgötter.

**Writing – original draft:** Minija Tamosiunaite, Christian Tetzlaff, Florentin Wörgötter.

**Writing – review & editing:** Minija Tamosiunaite, Florentin Wörgötter.

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
