## [Decision Letter · Decision Letter 0]

3 Sep 2023

Dear Dr. Tamosiunaite,

Thank you very much for submitting your manuscript "Unsupervised learning of perceptual feature combinations" (PCOMPBIOL-D-23-01087) for consideration at PLOS Computational Biology. As with all papers peer reviewed by the journal, your manuscript was reviewed by members of the editorial board and by several independent peer reviewers. Based on the reports, we regret to inform you that we will not be pursuing this manuscript for publication at PLOS Computational Biology. In particular, as highlighted by both reviewers, it is simply not clear that this work represents an advance on existing models. 

The reviews are attached below this email, and we hope you will find them helpful if you decide to revise the manuscript for submission elsewhere. We are sorry that we cannot be more positive on this occasion. We very much appreciate your wish to present your work in one of PLOS's Open Access publications. 

Thank you for your support, and we hope that you will consider PLOS Computational Biology for other submissions in the future.

Sincerely,

Daniel Bush

Academic Editor

PLOS Computational Biology

Lyle Graham

Section Editor

PLOS Computational Biology

Reviewer's Responses to Questions

**Comments to the Authors: **

Reviewer #1: The authors here analyze the general problem of detecting coincident patterns. In particular, they propose a pair of putative learning algorithms based on Hebbian principles with simulated annealing controlling the learning rate and stabilize the output of the cell. This acts to prevent the type o f run-away growth that Hebbian rules often lead to. However, detecting coincident patterns is easily achievable with a Perceptron learning rule and a multi-layer perceptron network. In fact, I would argue that the learning rules they authors consider are essentially slight modifications to the Perceptron learning algorithm where all patterns cause an increase in weights, up until they reach the annealing threshold. As such, the novelty here is minimal as time-varying learning rates in the perceptron learning algorithm are considered even as textbook exercises. It is rather unfortunate, as well, that Perceptrons and the Perceptron learning algorithm is never mentioned in this manuscript, despite the large similarity between both the problem the authors are trying to solve, and the methodology they use. I outline my points below. 

Major Concerns 

1) The specific problem these authors have considered is that of detecting coincident patterns. If we receive multi-modal stimuli that coincide, the neuron should respond but not to the stimuli themselves. If we consider the case of two stimuli, then we can define a binary vector [0,0], [0,1], [1,0], and [1,1] where 1 denotes the presence of a stimulus, and 0 denotes the absence. The goal then is to have a neuron respond exclusively to [1,1]. But, this is accomplished with Frank Rosenblatt's Perceptron learning algorithm, and perceptrons, simple computational units that threshold a linear combination of inputs. Note however that this is a convention. We can equally take the entry "0" to denote the presence of a stimulus, and 1 to denote its absence, or we can define any other vector set (e.g. [-1,-1],[-1,1],[1,-1],[1,1]) to denote the presence or absence of a stimulus. 

2) In fact, I think the authors have largely "rediscovered" the perceptron learning algorithm, and used it somewhat incorrectly. The author's themselves never cite Rosenblatt's original work, or other perceptron-like pattern classifiers that immediately solve this problem. I outline how the overall setup is largely identical to a perceptron learning algorithm point by point 

- The computational units have a set of inputs u, and a weight vector w. The computational units take the inner product of w with u and then apply some kind of thresholding. This is either smooth, as in the case of the sigmoid considered in equation (2) or discrete, as in equation (7) where y=w^T u must be thresholded by eta. A Perceptron is a computational unit that takes an input pattern x and applies a weight w as an inner product, and subsequently thresholds it:

y = 1 if w^T x >0, 0 if w^T x < 0

- The learning rules are largely perceptron-like. The perceptron learning rule alters the weights as 

w(t+delta) = w(t) + r(d_i-y_i)*x_i. 

where as the Hebbian rule alters the weight as 

w(t+delta )=w(t) + mu(t)*y_i*x_i

and the ALL rule alters the weight as.

w(t+delta) = w(t) + mu(t)*H(y_i-eta)*x_i 

where i is the index of the training sample. Let's consider the Hebbian rule, as it is a simpler rule to work with. For all 2D input patterns, [0,0],[0,1],[1,0],[1,1], the ALL-Rule returns the following weight changes 

Delta_W = [0,0] when x_i =[0,0]

Delta_W = [0,mu(t)] when x_i = [0,1] 

Delta_W = [mu(t),0] when x_i = [1,0] 

Delta_W = [mu(t),mu(t)] when x_i = [1,1] 

While the Perceptron learning rule would return the following 

Delta_W = [0,0]

Delta_W =[0,r(d_i-y_i)]

Delta_W = [r(d_i-y_i),0]

Delta_W = [r(d_i-y_i),r(d_i-y_i)]

Thus, if we consider a thresholded neuron where y_i is in the interval [0,1], and we set d_i = 1 to all the inputs, then we would largely obtain weight changes in a similar direction as the ALL rule. But this would imply all the patterns return a neural activation (y=1). 

The way the rule seems to work is that you just increase the weight corresponding to an input generally. In fact, this is shown in Figure 1. The weights always increase in response to an input pattern, ti's just the threshold is set to decrease the learning rate once a neuron fires. The coincidence detection is because the two inputs and two positive weights lead to an overall current that is larger than when just a single input is on. Normally however, in a Perceptron, you would have negative changes to the weights which allow you to set a boundary so that not all the responses have to be +1. The annealing is simply stopping the algorithm from yielding a "spiking" or threshold response to all possible input vectors, and so you obtain somewhat coincidentally pattern separation. 

Thus, the algorithms considered by the authors are 1) Perceptron-like in their deployment, 2) are likely to be less robust than a direct application of the Perceptron learning rule. This would also imply that the annealing threshold is always hand-tuned to guarantee the correct coincident detection. 

Minor Concerns 

1) throughout the manuscripts, the authors have type set the quotation marks wrong. In order to do the left quotation in Latex, one needs to press the ' key twice, while " is used exclusively for the right quotation 

2) This also occurs when using the single quote ' on line 32 for 'large enough' 

3) Munro vs Munroe in the BCM rule, on line 23. 

4) There are many statements made by the authors without supporting references. For example "annealing is usually applied as an additional mechanism to ensure an efficient convergence of weights"...examples? 

5) The figure legends are very spartan at times. For example, for Figure 2, the legend consists of 4 words, with little other description. 

6) I would advise the authors to put their code on a database with a referee read-only access password. I believe github and modeldb have such options.

Reviewer #2: This article addresses the issue of unsupervised development of an effective coding scheme for feature combinations, an issue that is important for various aspects of cognitive processing.

In my estimation, the most important element of this article is Figure 7B, bottom-right panel. This is a somewhat non-obvious result, in that it shows that the rule proposed by the authors is potentially capable of generating an ordered representation of feature combinations beyond pairwise, such that it lends itself to easy read-out by subsequent stages. I found it puzzling that this result is relegated to a tiny subpanel within a figure panel, and that it receives little attention beyond this demonstration. The authors should have expanded much more on this result, with more detailed investigations of the minimal conditions that are required for its emergence, its dependence on the number of inputs, parameterization, and others.

As for the remaining part of the article, which forms the bulk of the paper and which focuses almost entirely on pairwise combinations, I was not entirely convinced that it adds much on top of existing schemes. Let's focus on the Intrator-Cooper BCM rule, labelled BCM-rule by the authors. This is the only competing rule selected by the authors that appears as a valid contender to the ALL-rule proposed by the authors. Indeed, except for the result in Figure 7B mentioned above, the BCM-rule is able to accommodate pretty much all other results achieved by the ALL-rule (see for example Figure 7A).

When I look at equation 9 (BCM-rule), it seems plausible that it may be made to behave similarly to equation 3 (ALL-rule). \\mu and \\mathbf{u} are common to both rules. We are left with a bunch of stuff in equation 9 that depends on v, where v is a function of the linear output y, corresponding to g in equation 3, which is a thresholded (Heaviside) function of y. As noted by the authors, v can be made to approximate the thresholding function in equation 3 (lines 127-128), so the question is whether v^2*dv/dy (with \\Theta_{M} set to ~0) may behave like v when v is very sharp, which is not implausible depending on details. I may have missed it, but I could not find a detailed description of the parameterization adopted for the BCM-rule, so I assume that the authors chose a parameterization for v that is quite gradual. I wonder how the BCM-rule behaves for the specific case shown in Figure 7B when it is parameterized to implement a sharp thresholding function v. It may produce a better separation for the triple conjunction, for example.

If the Intrator-Cooper BCM rule can be parameterized to encompass the results achieved by the ALL-rule, it is still interesting to show that it can be significantly simplified to acquire unexpected properties like Figure 7 (bottom-right panel), but it does subtract somewhat from the contribution of this study in terms of novelty. The issue here is that the ALL-rule may bring some benefit in relation to a specific problem like the one examined here, but may underperform with respect to a different problem. If the BCM-rule can potentially cover the ALL-rule and also deal with other problems, it would remain a more attractive option.

Another (minor) criticism is that the authors choose relatively narrow ranges for input parameterization, for example they explore two inputs that differ in intensity by about 50%, which is not very useful for real biological processing: under real-world scenarios, features must be combined under much more extreme changes in input intensity. I realize that the approach can be augmented to incorporate normalization schemes that act early, but it is not then clear that the ALL-rule would preserve the ordered structure demonstrated in Figure 7B under those conditions. I think this aspect of the study needs further investigation/clarification.

An issue related to the one above is that, possibly as a consequence of the limited amount of details provided by the authors in relation to the training procedure, it was not clear to me how robust the ALL-rule is to potential differences between the training cohort and a subsequent testing cohort that differs significantly, particularly in relation to coding stability. For example, once the network is frozen, is it stable when coding input distributions that differ from those used during training, and that differ among themselves? By stable I mean that if a given neuron codes for combination 13 or 123, it retains its label regardless of substantial changes in the parameterization of the input. For example, if input 1 was twice as intense as input 2 during training, and the network is characterized for this configuration, when I then characterize for a configuration in which input 1 is half as intense as input 2, is the code stable? These issues do not seem to be adequately addressed.

**Have the authors made all data and (if applicable) computational code underlying the findings in their manuscript fully available?**

Reviewer #1: No: The authors won't make their code public until after the manuscript has been accepted, but have also not included links for a referee.

Reviewer #2: None

PLOS authors have the option to publish the peer review history of their article (what does this mean?). If published, this will include your full peer review and any attached files.

Reviewer #1: No

Reviewer #2: No

---

## [Decision Letter · Decision Letter 1]

18 Jan 2024

Dear Dr. Tamosiunaite,

Thank you very much for submitting your manuscript "Unsupervised learning of perceptual feature combinations" for consideration at PLOS Computational Biology. As with all papers reviewed by the journal, your manuscript was reviewed by members of the editorial board and by several independent reviewers. The reviewers appreciated the attention to an important topic. Based on the reviews, we are likely to accept this manuscript for publication, providing that you modify the manuscript according to the review recommendations.

In particular, the authors should do a little more to clarify the distinction between their unsupervised learning rule and the Perceptron rule, as well as the advantages of taking an unsupervised approach (i.e. biological plausibility, no need for feedback). However, I do not think they need to formally compare the results obtained using each approach.

Sincerely,

Daniel Bush

Academic Editor

PLOS Computational Biology

Lyle Graham

Section Editor

PLOS Computational Biology

Reviewer's Responses to Questions

**Comments to the Authors:**

Reviewer #2: I should state upfront that, in general, I do not like to re-review rejected papers. This kind of situation makes peer review messy. Having said that, I sympathize with the authors in that they seem to have reasonable grounds for appeal in this case.

I should also notice upfront that, when going back to this manuscript, I was irritated by the fact that the authors did not highlight revisions using, for example, red ink. This means that Reviewers must re-read the paper from scratch, which adds to their workload (and I am reviewing another 4 papers on top of this one). In the future, please be more respectful of Reviewers' time by clearly highlighting portions of the manuscript that have been modified during the revision process.

Now, with all that in mind, I am trying to keep an open mind here. The authors have expanded on the feature I singled out during the first round, namely the result in Figure 7 that was originally relegated to one subpanel. In my opinion, that is the entire paper: without that, I would not consider this manuscript above threshold for PLOS CB. With that result, I think it deserves consideration. In my understanding, it is an interesting result that such a simple rule can generate an ordered representation of feature combinations that is essentially transparent for further read-out by higher modules.

I was somewhat disappointed by the fact that the authors did not take on the challenge I set out for them in the first round regarding stability of the coding strategy (point they refer to as Q6 in their rebuttal). At the same time, I accept that this is a very challenging problem, so I am willing to concede on that point and accept that it will require further research beyond this paper to be fully addressed.

All in all, the authors have addressed my comments to a reasonable degree of satisfaction.

Reviewer #3: I will not provide a detailed review of this paper, as I was brought in after an initial round of reviews had already taken place. Instead, I will focus primarily on the dispute/appeal regarding reviewer 1's comments on the manuscript. Having read through the reviewer comments and author responses, I am satisfied that the author's are correct in their response, and that the reviewer's objection was based on a misconception/failure to distinguish between supervised/unsupervised learning rules. That said, the very fact that the reviewer (who is presumably an expert in the field) was able to come to this erroneous conclusion suggests that the authors have not taken enough care to clarify the essential contributions of their study and the relevance to previous work. Therefore, I would recommend that the authors explicitly discuss the distinction between supervised/unsupervised learning rules, and make clear to the reader how their work goes beyond simple perceptron learning rules/is more biologically plausible, etc. (it is of course up to the authors to decide how to frame the contributions of their paper). While the authors have added a small paragraph attempting to address this, I found this unsatisfactory and somewhat dismissive of reviewer 1's concerns (e.g., "these two approaches cannot really be compared"). Instead, the authors should explain why this distinction/advance is important, and why this makes the proposed learning rule advantageous. Even better would be if the authors could formally (via numerical simulations and/or mathematical analysis) compare and contrast the supervised perceptron learning rule vs their method (of course supervised will do better, and that's not a problem, but some general insight into how the two learning rules relate could be interesting). Overall, the comparison of this learning rule to the classic perceptron learning rule seems important, as has not been sufficiently elaborated in the manuscript as far as I can tell.

**Have the authors made all data and (if applicable) computational code underlying the findings in their manuscript fully available?**

Reviewer #2: Yes

Reviewer #3: Yes

PLOS authors have the option to publish the peer review history of their article (what does this mean?). If published, this will include your full peer review and any attached files.

Reviewer #2: **Yes: **Peter Neri

Reviewer #3: No

Figure Files:

Data Requirements:

Reproducibility:

References:

---

## [Decision Letter · Decision Letter 2]

19 Feb 2024

Dear Dr. Tamosiunaite,

We are pleased to inform you that your manuscript 'Unsupervised learning of perceptual feature combinations' has been provisionally accepted for publication in PLOS Computational Biology.

Best regards,

Daniel Bush

Academic Editor

PLOS Computational Biology

Lyle Graham

Section Editor

PLOS Computational Biology

Reviewer's Responses to Questions

**Comments to the Authors:**

Reviewer #2: NA

Reviewer #3: The authors have addressed my comments.

**Have the authors made all data and (if applicable) computational code underlying the findings in their manuscript fully available?**

Reviewer #2: None

Reviewer #3: Yes

PLOS authors have the option to publish the peer review history of their article (what does this mean?). If published, this will include your full peer review and any attached files.

Reviewer #2: **Yes: **Peter Neri

Reviewer #3: No

---

## [Editor Report · Acceptance letter]

27 Feb 2024

PCOMPBIOL-D-23-01087R2 

Unsupervised learning of perceptual feature combinations

Dear Dr Tamosiunaite,

I am pleased to inform you that your manuscript has been formally accepted for publication in PLOS Computational Biology. Your manuscript is now with our production department and you will be notified of the publication date in due course.

With kind regards,

Anita Estes
